# Preparatory phase of large earthquakes illuminated by unsupervised categorization of earthquake catalog features

Sadegh Karimpouli [1] ✉, Patricia Martínez-Garzón [1,2], Sebastián Núñez-Jara[1], Matteo Picozzi [3], Daniele Spallarossa[4], Grzegorz Kwiatek[5], Georg Dresen [1,6], Marco Bohnhoff[1,7] & Gregory C. Beroza [8]

Predicting large earthquakes remains a significant challenge due to the complexity of fault systems and the variability of preparatory processes. We introduce an unsupervised machine learning framework to categorize seismicity patterns and identify, when present, seismicity transients preceding large earthquakes. We focus on five large earthquakes and extract seismomechanical features per families of events, defined as clustered events in space, time and magnitude. Here we show that for those cases displaying a preparatory phase, specific long-lasting families belonging to a critical category signalling an upcoming earthquake occur during the preparatory phase. Compared to other periods, critical categories reflect a higher spatial-temporal localization, earthquake interaction and strain release. The method will not detect such a transient for earthquakes with no detectable seismic preparatory phase. Finally, we demonstrate that the method is capable of identifying preparatory phases (when present), showing potential for operational earthquake forecasting.

Predicting the timing, location and size of future earthquakes remains a long-standing and unresolved - if not impossible - challenge in geosciences. For decades, scientists from different disciplines have aimed to identify precursors, denoting specific preparatory patterns that may occur before large earthquakes[1]. The most common earthquake preparatory processes include foreshocks and slow slip transients, which may occur directly around the future mainshock epicentre or in their vicinity. Observations along diverse types of faults and plate boundaries have revealed significant variability, ranging from precursory signals on various spatio-temporal scales to no detectable preparatory phase[2,3]. If these differences between precursory observations from different sequences are not merely due to differences in the monitoring networks, the observed variability may originate from the inherent complexity of the earthquake nucleation

process[4]. This complexity is likely due to variations in geological materials and structures (e.g., rock/fluid properties and fault zone characteristics) as well as differences in tectonic loading conditions.

The underlying physical processes that control the run-up to failure have been studied through theoretical models[5], numerical simulations[6], laboratory experiments[7], and the comparison of their results with well-documented field observations[8–10]. Previous studies have highlighted that the earthquake preparatory processes can involve various physical mechanisms occurring on different temporal and spatial scales resulting in the build-up of stress and elastic strain on a fault[4,7]. Before rupture, distributed damage progressively localizes reducing fault strength[11]. This may lead to coalescence of fault segments and enhanced earthquake interaction[12]. Stress correlations at large wavelengths may facilitate earthquake propagation and result in

[1]GFZ Helmholtz Center for Geosciences, Potsdam, Germany. [2]RWTH University of Aachen, Aachen, Germany. [3]National Institute of Oceanography and Applied Geophysics – OGS, Trieste, Italy. [4]DISTAV, University of Genoa, Genoa, Italy. [5]GMuG Gesellschaft für Materialprüfung und Geophysik mbH, Dieselstraße 9, Bad Nauheim, Germany. [6]University of Potsdam, Potsdam, Germany. [7]Department of Earth Sciences, Freie Universität Berlin, Berlin, Germany. [8]Department of Geophysics, Stanford University, Stanford, CA, USA. ✉e-mail: sadegh.karimpouli@gfz.de

larger earthquakes[13]. Increased fault roughness, thought to be present in immature fault structures and heterogeneous plate boundaries tends to promote an interplay between seismic and aseismic slip, which in turn, may favour the occurrence of foreshocks and slow slip

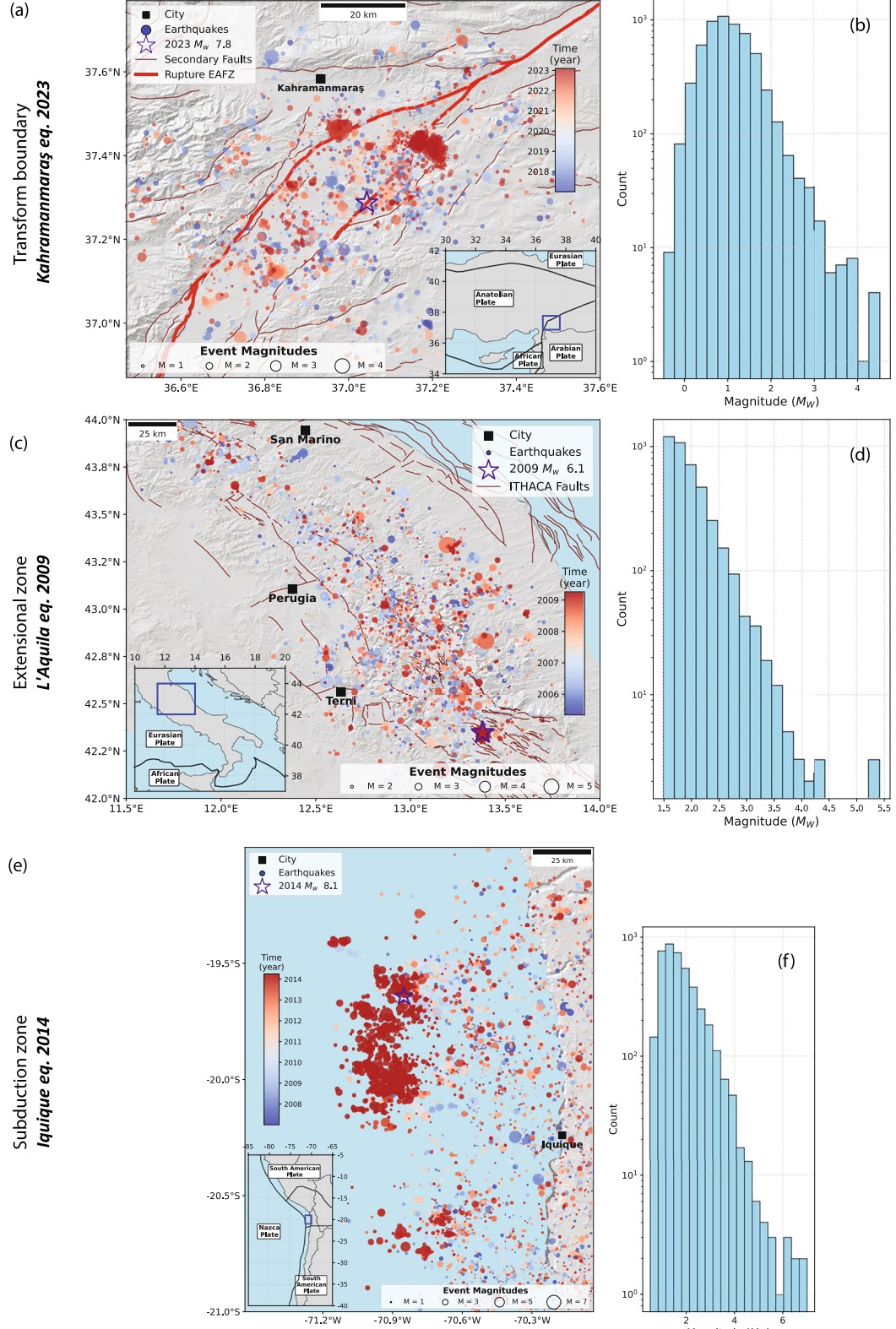

**Fig. 1 | Seismicity and magnitude distributions preceding three major earthquakes with known preparatory phase. a, b** 6 years before 2023 MW 7.8 Kahramanmaraş, (**c, d**) 4 years before 2009 MW 6.1 L'Aquila and (**e, f**) 7 years before 2014 MW 8.1 Iquique earthquakes. Events are shown in circles, sized by magnitude and color-coded by time. Mainshock epicenters are marked with stars. Faults are shown in brown thin lines, while rupture along the EAFZ is shown in thick red line, in the case of Kahramanmaraş (**a**). ITHACA: ITaly HAzards from CApable faulting (Dataset is from the portal of the geological survey of Italy). Insets show location of maps.

events preceding mainshocks[14]. Migration of fluids along faults may add an extra layer of complexity, with elevated pore pressures typically promoting the occurrence of dynamic ruptures[15,16].

Identifying preparatory processes of upcoming large earthquakes involves the continuous monitoring of the stress and fault deformation through various seismological or multidisciplinary approaches. The joint analysis of these multi-parameter datasets can be based on the analysis of various seismo-mechanical and statistical features over different spatial and temporal windows[17–19]. Relevant examples of such features include the b-value, thought to be inversely related to the fault stress level[20] or level of damage in the rock mass[21], and properties reflecting the degree of earthquake interaction and localization, such as spatio-temporal earthquake clustering or correlation integral, and the cumulative seismicity rates and seismic release[22]. Correlated variation of these features, for example, combined b-value decrease, increase in seismicity rate and progressive spatio-temporal localization, may reveal a transient from stable (background) seismicity (Poissonian distribution) to a critical stress state of the fault close to failure, in cases leading to large earthquakes[23]. In this study, we refer to the term 'critical' to describe families of events with peculiar characteristics that retrospectively occur in the proximity to failure.

The advent of Machine Learning (ML) data-driven approaches represents a potential paradigm shift in improving earthquake forecasting. In some laboratory rock deformation experiments involving a large monitoring data sets from simple and well-characterized (smooth) fault conditions, the time-to-failure has successfully been predicted[24] employing supervised approaches. However, unsupervised techniques, where diagnostic features are not predefined but are discovered in the data, offer an alternative approach to this problem[25,26]. Several pioneering approaches have already demonstrated the effectiveness of this exploratory approach to discern the preparatory stages of landslides[27] and volcanic eruptions[28]. Analysing stick-slip experiments on granite samples containing a rough fault (hence containing larger complexity than previous studies), Kwiatek et al.[7] extracted a pool of catalog-driven features reflecting the state of stress in the lab fault. They revealed a transition from stable deformation to an intermittent critical state that promoted the occurrence of large events. Karimpouli et al.[29] used unsupervised learning to categorize these features that reflected the temporal evolution of the state of stress.

Can the methodologies derived for laboratory earthquakes be upscaled to monitor the state of stress of tectonic faults and hence identify earthquake preparatory processes (when present)? Experimental work represents a unique opportunity to examine in detail the methods and conditions that promote detectability of preparatory processes. However, upscaling is challenging, mainly due to the increased complexity of natural faults with respect to laboratory conditions, as described above. In addition, in the field, monitoring resolution is limited and material and structural properties, fault roughness or the the presence of fluids, are poorly known. So, what are the best approaches to monitor and successfully identify earthquake preparatory processes (if present), while minimizing false alarms? In this study, we expand our feature computation and unsupervised methodology originally developed for labquakes[30] and test its performance when upscaled and applied to large earthquakes. We focus on three earthquake sequences occurring in different tectonic settings that were preceded by seismicity transients: the 2023 $M_W$ 7.8 Kahramanmaraş earthquake along the transform boundary of the East Anatolian Fault Zone in Türkiye, the 2009 $M_W$ 6.1 L'Aquila, in the extensional zone of central Italy and the 2014 8.1 $M_W$ Iquique earthquake, in the Chilean subduction zone. For these earthquakes, previous studies have documented months-to-weeks long seismicity transients around the future epicentral or rupture areas[9,31,32]. Our methodology successfully identifies such transients, interpreted as the preparatory phase of the forthcoming large events. We then present two more challenging cases: i. The 2016 $M_W$ 6.2 Amatrice earthquake in

the central Apennines in Italy, where the preparatory phase is not reflected in the seismicity catalog, and ii. The 2024 $M_W$ 7.5 Noto earthquake, Japan, where long-lasting swarm activity is known as the main preparatory signature. In both cases, our workflow extracts dominant seismicity patterns defining discrete categories. In contrast to the other large earthquakes considered, the seismicity before these two mainshocks does not show critical characteristics.

In this work, we monitor the characteristics of families of events during the earthquake preparatory phase. We define a family as a cluster of events with a nearest neighbour time-space-magnitude distance[33] smaller than a threshold. The key ingredients of our workflow are: i) the computation of event-based catalog features in various space and time windows[30], ii) the separation between background and clustered seismicity with a nearest neighbour method[12] and the identification of families, iii) the introduction of per-family features by assigning an average feature values to each family, and iv) the categorization of these families according to the features found through unsupervised learning. To avoid confusion between two distinct uses of the term 'clustering' namely, 'seismic clustering'[34] and 'clustering algorithms'[35], we refer to the latter as 'categorization' and use 'category' instead of 'cluster' in this context. By focusing on families, we emphasize the processes and seismicity highlighting the earthquake interaction, which, according to available observations and models, tends to increase on the run-up to the mainshock[2,4]. We demonstrate that, in the 2023 $M_W$ 7.8 Kahramanmaraş and $M_W$ 6.1 L'Aquila and the 2014 8.1 $M_W$ Iquique earthquakes, our framework effectively identifies a category of seismicity before the mainshocks exhibiting anomalous properties relative to the other previous categories. In the cases of 2016 $M_W$ 6.2 Amatrice and 2024 $M_W$ 7.5 Noto earthquakes, where there is no detectable preparatory phase, no anomalous family is detected. These results demonstrate the capability of this method to be engaged to an operational earthquake forecasting system.

## Results
### The case studies
**Kahramanmaraş.** On February 6, 2023, at 01:17 UTC, a catastrophic $M_W$ 7.8 earthquake struck south-eastern Türkiye, with its epicentre near the city of Kahramanmaraş (Fig. 1a)[36]. This event occurred on the East Anatolian Fault Zone (EAFZ), which forms the boundary between the Anatolian and Arabian tectonic plates. The rupture extended across multiple fault segments, reaching the surface and covering approximately 500 km in length[37]. Just nine hours later, a $M_W$ 7.5 Elbistan earthquake struck about 90 km to the north-northwest, near Ekinözü, Türkiye[36].

The area up to 65 km from the epicentre of the 2023 $M_W$ 7.8 Kahramanmaraş earthquake displayed prominent seismic activity, b-value decrease, and enhanced earthquake localization and interaction, which began approximately eight months before the mainshock[9,38]. Based on recordings from the regional seismic network operated by the Turkish Disaster and Emergency Management Presidency (AFAD) and the Kandili Observatory Network (KOERI), Núñez-Jara et al.[39] developed a high-resolution enhanced seismicity catalog on the mainshock epicentre region from 2017 until the 2023 mainshock, containing 5,686 earthquakes within a radius 50 km from the mainshock epicentre. The catalog displays a magnitude of completeness of $M_C = 1.0$ (Fig. 1b).

**L'Aquila.** The 2009 L'Aquila earthquake sequence was a destructive seismic event that struck the Abruzzo region, central Italy, with a $M_W$ 6.1 mainshock on April 6, 2009 (Fig. 1c). This sequence included five significant shocks ($5.0 \leq M_W \leq 6.1$)[40], primarily occurring during the first two weeks of April 2009. The earthquakes activated an area spanning approximately 50 km along the central Apennines. The foreshock sequence preceding the L'Aquila mainshock lasted several months, and involved the main fault plane and adjacent fault

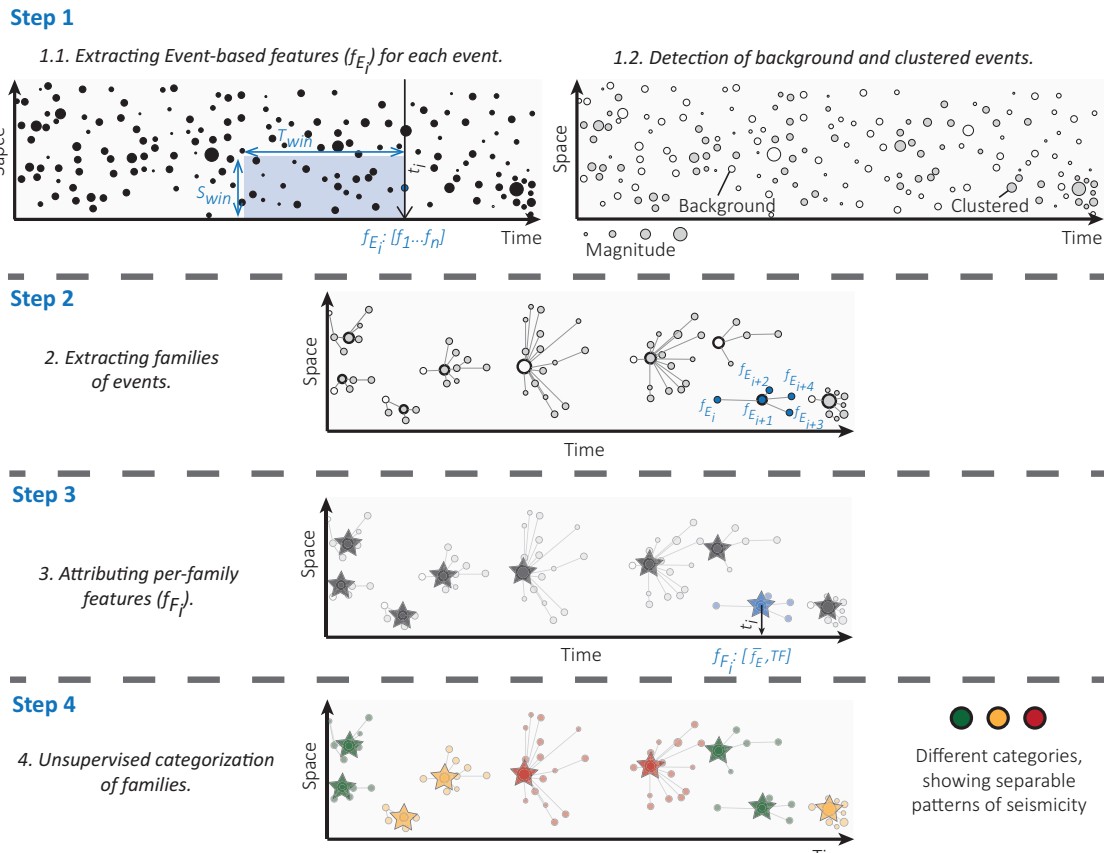

**Fig. 2 | Flowchart of the proposed method, showing a seismicity catalog in time and space, where the size of each event (circle) represents its magnitude.** (Step 1.1) Event-based features ($f_{E_i}$) are computed based on the events inside spatial ($S_{win}$) and temporal ($T_{win}$) windows and are assigned to each event in its time ($t_i$) (see Methods and Table S1), (Step 1.2) all events are classified as either background (empty circles) or clustered seismicity (filled circles) following Zaliapin and Ben-Zion[39]. (Step 2) We identify earthquake families from clustered seismicity by linking events according to their nearest-neighbour distances after removing background seismicity. Each family consists of a mainshock (the largest-magnitude event within the family, shown with bold circles) and a foreshock–aftershock sequence (events occurring before-after the mainshock). Note that each event in the family is attributed a vector of n event-based features ($f_{E_i}$) from step 1.1. (Step 3) Per-family features ($f_{F_i}$) are derived for the mainshock time (represented by stars) by computing $\overline{f_E}$: the average values of all event-based features plus TF: topological features. (Step 4) K-means algorithm is used to categorize all families via unsupervised learning. Blue colored circles and stars, in steps 2 and 3, show an example of $i$th event and family. Green, orange and red colors show different categories of seismicity families.

segments[41]. Picozzi et al.[22] reported persistent anomalies in the Energy Index (Ei) in the years following mainshock, including high-Ei patches near the future Amatrice hypocentre, suggesting a long-term evolution of crustal stress conditions. The 2009 $M_W$ 6.2 L'Aquila earthquake ruptured a set of normal faults parallel to the trend of the Apennine Mountains. The rupture area displayed abundant seismicity during the three months preceding the mainshock[31,42].

The catalog data is obtained from the RAMONES service, which provides a fully automated and robust processing strategy for central Italy[43]. The catalog is enhanced not in the sense of including very low-magnitude earthquakes, but rather because it provides independent estimates of seismic moment and radiated seismic energy, derived directly from waveform measurements rather than empirical magnitude-energy relationships. The RAMONES catalog includes 4,075 events with magnitude $M_C = 1.5$ (Fig. 1d), recorded between 2005 and the time of the mainshock, covering central Italy and adjacent areas.

**Iquique.** The 2014 $M_W$ 8.1 Iquique is among the best-documented megathrust earthquakes in subduction zones, owing to extensive seismological and geodetic observations that captured its run-up, rupture onset, and the postseismic response. The mainshock was preceded by an extended preparatory phase, beginning about eight months earlier, characterized by interacting seismic and aseismic

transients. During this period, bursts of seismicity and multiple slow-slip events were observed, the most prominent of them starting with an upper-plate $M_W$ 6.7 foreshock two weeks before the mainshock[32,44]. Statistical analyses of the background seismicity and $b$-value patterns further revealed a years-long, progressive weakening of the future rupture area, setting the stage for the rupture of the main asperity[45,46].

The Integrated Plate Boundary Observatory (IPOC) catalog is a semi-automatically compiled catalog covering 2007-2021 in Northern Chile[47]. It is built upon data from IPOC, the Centro Sismológico Nacional (CSN), GEOFON, and several temporary networks. For our analysis, we use the portion of the catalog spanning 2007 to 1 April 2014 (the day of the mainshock), covering the inter-plate and upper-plate seismicity around the eventual rupture zone. This subset contains 4134 events with a magnitude of completeness $M_C = 1.7$, and includes numerous $M_W > 5$ earthquakes, among them two $M_W > 6$ events in 2008 and 2009 that did not culminate in a major rupture.

## Categorizing seismicity families using unsupervised learning
To test whether we can capture the characteristics evolving during the preparatory phase of these three mainshocks, we focus on the evolution of properties within seismicity families rather than analysing individual events. We compute the per-family features as follows (Fig. 2):

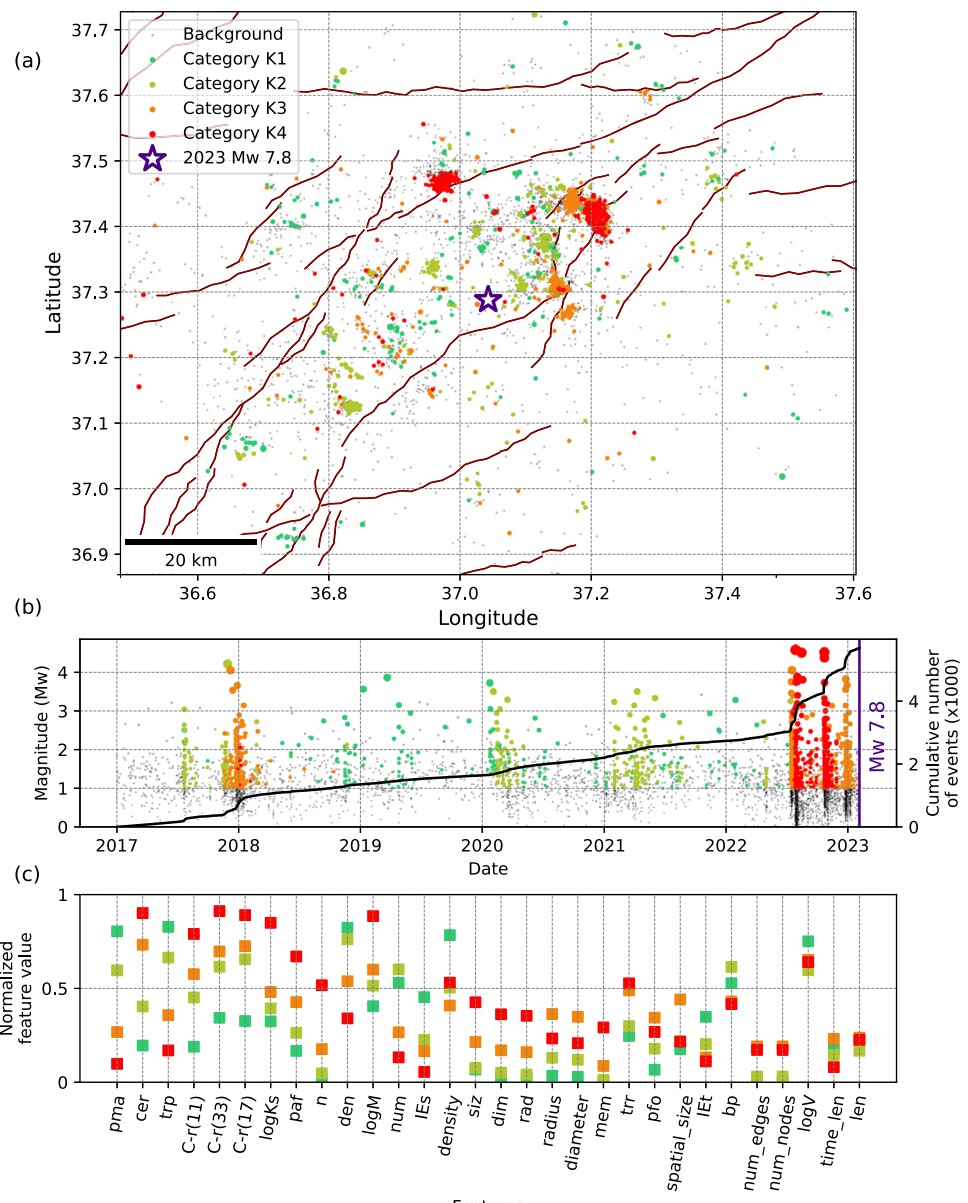

**Fig. 3 | Categorization of seismicity families prior to the 2023 MW 7.8 Kahramanmaraş earthquake. a** Map view showing the spatial distribution of event family members color-coded with their corresponding category. Background events are shown in grey. **b** Magnitude-time distribution of event family members, with the cumulative number of all (background and clustered) events represented by solid lines. **c** Feature values at the centroid of each category, sorted from highest to lowest separability. To improve visualization, we use an average value of each feature type over 4 spatial and temporal windows (2×2) and show only one value per feature type, leading to 30 (23+7) features. The colour scheme reflects the evolution of families, transitioning from a stable state (green) to a critical state (red). Description and explanations of individual features are provided in Table S1.

**Step 1.1**. We compute and assign event-based features to each event (Fig. 2, also see Methods). To this end, we use 23 different features (Table S1). These quantify general characteristics of the seismicity (event/moment rate, b-value), localization in time, space, time-space and time-space-magnitude, earthquake interaction and properties of event families. Based on two spatial and temporal windows (see Methods and Table S2), this leads to a pool of 92 features (23x2x2). However, for the L'Aquila earthquake, we also have access to the Energy Index[22] and, therefore, we compute 96 features (24x2x2).

**Step 1.2**. The background and clustered events are identified based on nearest-neighbour distance and clustering analysis[33] (Fig. 2). This provides the basis for the extraction of event families.

**Step 2**. A *family* is defined as a sequence of events connected by strong links based on nearest-neighbour distances below a

threshold value (Fig. 2). We identify and extract 96, 186 and 44 different event families for the 2023 $M_W$ 7.8 Kahramanmaraş, 2009 $M_W$ 6.1 L'Aquila and 2014 $M_W$ 8.1 Iquique earthquakes, respectively (see Figs. S1-S6).

**Step 3**. Per-family features are computed with two sub-sets of the features (Fig. 2): a) The average of all event-based features for each family (92, 96 and 92 features for Kahramanmaraş, L'Aquila and Iquique, respectively), and b) A set of 7 topological characteristics of the family such as the number of events, connection patterns, and other graph characteristics, that are independent of space and time windows (see bold features in Table S1). Therefore, a pool of 99, 103 and 99 features in total are computed for each family included in the catalogs of seismicity preceding the Kahramanmaraş, L'Aquila and Iquique earthquakes, respectively.

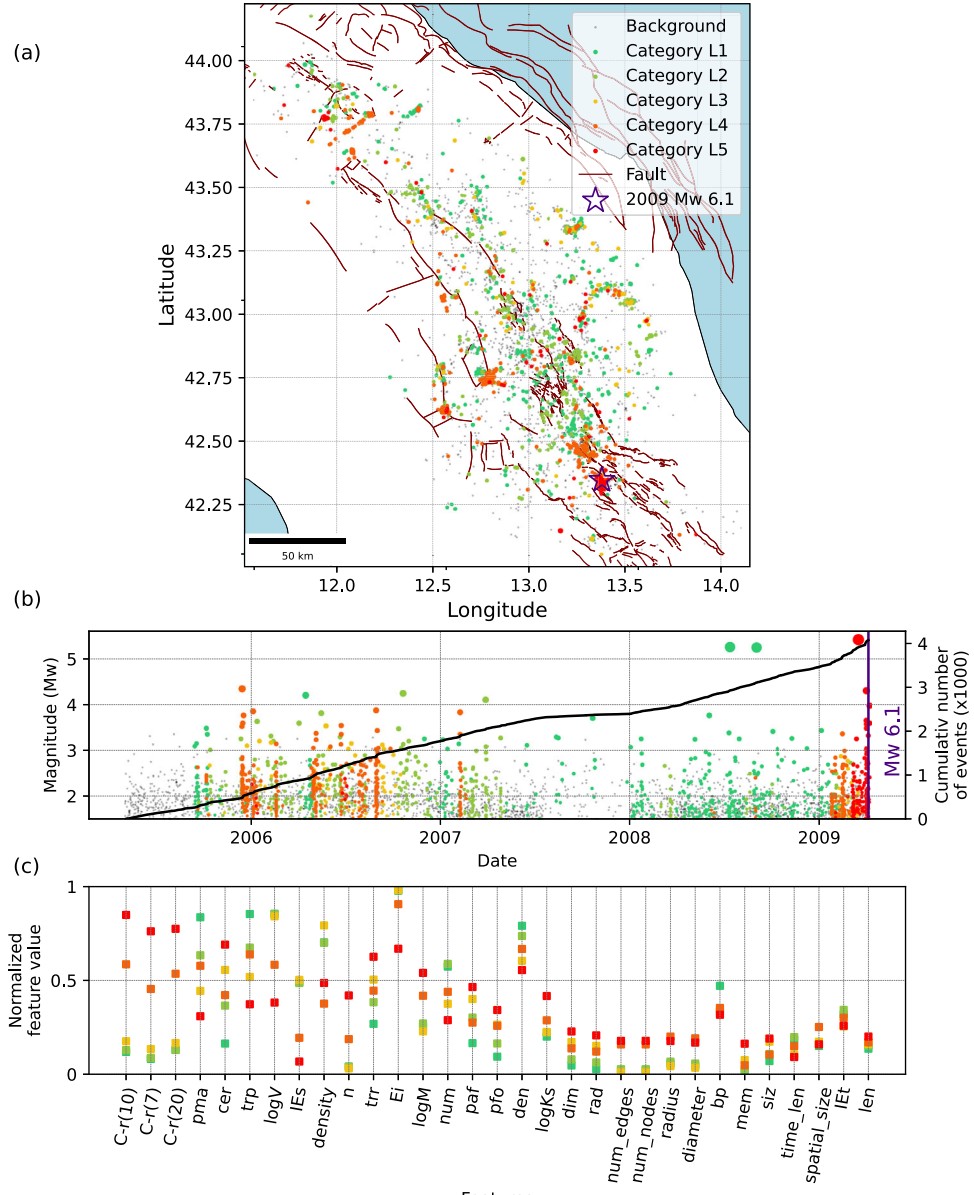

**Fig. 4 | Categorization of seismicity families prior to the 2009 MW 6.1 L'Aquila earthquake. a** Map view showing the spatial distribution of event family members color-coded with their corresponding category. Background events are shown in grey. **b** Magnitude-time distribution of event family members, with the cumulative number of all (background and clustered) events represented by solid lines. **c** Feature values at the centroid of each category, sorted from highest to lowest

separability. To improve visualization, we use an average value of each feature type over 4 spatial and temporal windows (2×2) and show only one value per feature type, leading to 31 (24+7) features. The colour scheme reflects the evolution of families, transitioning from a stable state (green) to a critical state (red). Description and explanations of individual features are provided in Table S1.

**Step 4**. We use per-family features as inputs for the K-means algorithm to categorize families in an unsupervised manner[34] (Fig. 2). This allows finding different seismicity patterns based on physics-informed features extracted from the seismicity catalog.

Note that we also tested alternative clustering algorithms, including spectral clustering, 'Ward' hierarchical clustering, Gaussian Mixture Models (GMM), and Density-Based Spatial Clustering of Applications with Noise (DBSCAN). The results for the Kahramanmaraş earthquake are presented in Supplementary Figs. 7–10, showing that only K-means and GMM produced coherent and physically meaningful categories that align well with the observed seismicity history in this region. Given the simplicity, interpretability, and computational efficiency of K-means, we employed this algorithm for all cases analysed in this study. Our goal is to assess how similar these families are and

determine which families belong to the same category. Since the number of categories is not predefined, we determine the optimal number of categories using the within-category sum-of-squares (WCSS) criterion, which is a fundamental distance measure in K-means clustering. WCSS is calculated by finding the squared Euclidean distance between each data point and its category's centroid, then adding up these values for all categories. Our investigation shows that this is the most reliable criterion for this study (see Supplementary Fig. 11). This yields four, five and four optimal categories for the Kahramanmaraş (K1-K4), L'Aquila (L1-L5) and Iquique (I1-I4) earthquakes, composed of different families, respectively.

To evaluate the occurrence of swarm-like activity, we use topological features of the families. Following Zaliapin and Ben-Zion[48], a topological average leaf depth can characterize the sequence, where

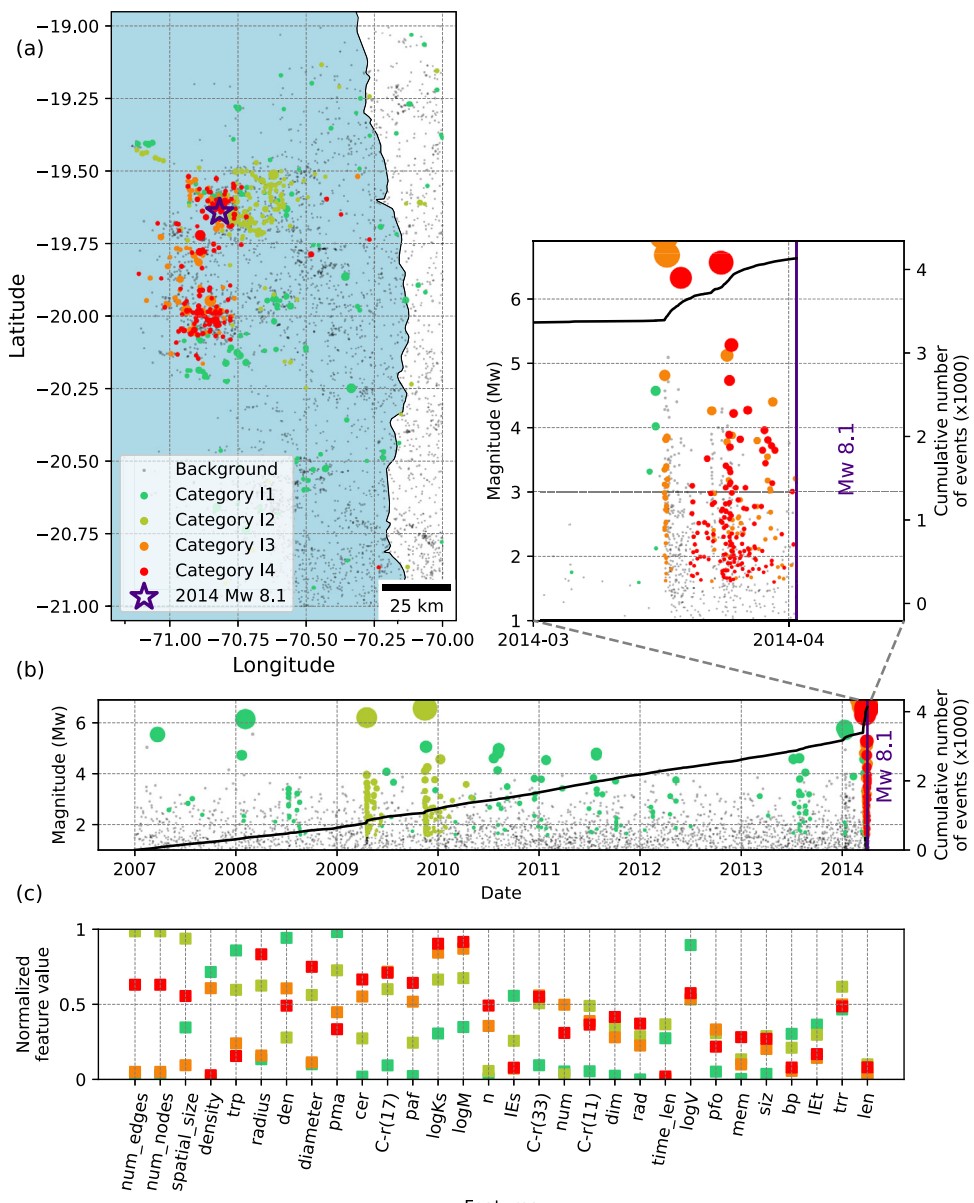

**Fig. 5 | Categorization of seismicity families prior to the 2014 MW 8.1 Iquique earthquake. a** Map view showing the spatial distribution of event family members color-coded with their corresponding category. Background events are shown in grey. **b** Magnitude-time distribution of event family members, with the cumulative number of all (background and clustered) events represented by solid lines. Top-right is the zoom view of the last two weeks before the mainshock. **c** Feature values at the centroid of each category, sorted from highest to lowest separability. To improve visualization, we use an average value of each feature type over 4 spatial and temporal windows (2×2) and show only one value per feature type, leading to 30 (23+7) features. The colour scheme reflects the evolution of families, transitioning from a stable state (green) to a critical state (red). Description and explanations of individual features are provided in Table S1.

the leaf depth counts the number of links from the first event to each leaf and a leaf is an event with no offspring. Similarly, we use additional graph-based properties to assess whether a family topology is chain-like (swarm) or umbrella-like (mainshock–aftershock or burst). For example, the density measures the number of connections relative to the maximum possible, where lower density indicates more swarm-like topology. Other measurements such as radius and diameter are derived from the eccentricity of the events defining the family. The radius/diameter are defined as the longest shortest path from a member to any other member, with the radius/diameter being the minimum/maximum eccentricity values. Accordingly, higher values of both metrics correspond to more swarm-like activity.

We present the results of our categorization of families in Figs. 3–5. The K-means algorithm categorizes families arbitrarily, meaning it does not consider the criticality of each category, with

criticality describing the proximity to mainshock[49]. By leveraging the physics-informed features, we analyse how each of the employed features differs among categories (Figs. 3–5c). Intuitively, only features that differ significantly between categories are expected to inform on the state of each category. Therefore, we sort the categories according to the feature values. To achieve this, we use the feature values corresponding to the centroid of each category. The central point in one category is defined based on the minimum sum of square distance relative to all points in the same category. This point is coordinated with 'm' dimensions, where 'm' is the number of features (m = 99, 103 and 99 per-family features for Kahramanmaraş, L'Aquila and Iquique earthquakes, respectively). Figures 3–5c visualize the feature space using normalized values of the centroid of each category. These plots offer a clear description of the feature values within each category varying between 0 and 1. This illustrates how each feature contributes

to the separation of the categories. We use the clustering (trp; pma; cer) and localization (C-r) features as well as seismicity rate (n) and Kostrov strain (logKs) to determine how stable/critical each category is.

For the 2023 $M_W$ 7.8 Kahramanmaraş sequence, in addition to the seismicity transients observed eight months before the mainshock[9], a second time period in late 2017-early 2018 displays comparable seismicity features[39]. Our analysis shows that, while event families with similar properties (i.e., category K3 – orange colour) appear in both spatial and temporal intervals, category K4 (except for a small family in 2018) is mainly observed over a long duration through the eight months before the $M_W$ 7.8 Kahramanmaras earthquake (Fig. 3a and b). This category is characterized by exceptionally high temporal and spatial localization, along with a relatively larger Kostrov strain release, differentiating it from all other families.

For the 2009 $M_W$ 6.1 L'Aquila earthquake (Fig. 4), previous studies showed that the preparatory phase began approximately three months before the event[31,42]. This is clearly reflected in Fig. 4a and b, where two distinct categories, L4 and L5, are concurrently identified within the same time period. Category L4 appears in other locations and time periods, particularly in 2006-2007 as a dominant family. However, long-lasting families of category L5 emerge mainly before the mainshock (except for small families with short durations in 2006-2007). This category is also characterized by high localization and strong temporal clustering (Fig. 4c).

For the Iquique earthquake, highly critical categories emerge approximately two weeks before the mainshock. Although many studies reveal longer preparatory phases of months-long and even years-long progressive weakening[32,50], our results only reveal the last phase expressed in seismic activity. Ruiz et al.,[32] and Schurr et al.,[50] also explain a mixture of foreshock swarms and short slow-slip transients in the last two weeks. Accordingly, our method can identify two categories in this time span (zoom view in Fig. 5). The topological features of families included in the different categories reveal that category I4 represents more swarm-like activity compared to category I3.

The key finding of our analysis is that any changes in the seismicity pattern evolving during the earthquake preparatory process could be illuminated by the properties of seismicity families and separated in different categories in this workflow. Importantly, our approach allows identifying a set of the most important features and their values distinguishing between family categories (Figs. 3–5c). These features are:

i) Clustering features such as the event proximity and the proportion of mainshocks to fore and aftershocks, which are observed to decrease, corresponding to the higher productivity of the sequences and the spatio-temporal localization.

ii) The correlation integral, which is observed to increase in agreement with the observed spatial point-localization of the event families.

iii) Kostrov strain, quantifying the enhanced seismic strain release, which is observed to increase during the preparatory phase.

iv) 3D volume and spatial interevent distance, which are observed to decrease signifying again the higher event localization.

Comparing the temporal evolution of the categories with the cumulative number of events (Fig. 3b, e) shows that notable shifts in the seismicity rates correspond to changes among categories. The workflow allows one to decide about the criticality state of each category based on a combined evolution of a high dimensional feature space. For each of the studied earthquakes, there are families particularly localized in space and time (categories K4, L5 and I4), which are sensitive to the preparatory process of seismic activity before a larger event (the criticality of the seismic activity). This suggests that the properties of clustered seismicity in the proximity of the mainshocks may be different from those at previous times.

Although a common, general behaviour is observed for the preparatory phases, note however that the relative importance of features may vary between different earthquake sequences.

## Discussion

In an effort to develop forecasting models, considerable work has been invested in designing and testing approaches that help to understand when faults might be approaching instability. For example, Keilis-Borok and his team developed algorithms to recognize patterns of statistical properties of the seismicity, which led, for some time, to claims of success[51,52].

Machine Learning has efficiently contributed to decrease the detection threshold of earthquake catalogs[53,54]. In some settings, ML has also allowed substantially improved earthquake forecasting and estimating fault displacement[55,56]. Popular approaches include the analysis of continuous waveforms from a single station[27,57,58] or the characteristics of an entire population of seismic events[29,59]. In the workflow proposed here, we exploit the fact that earthquakes tend to display enhanced interaction as rupture approaches[2,17]. We focus our analysis on the families of clustered events in the normalized space-time-magnitude domain. Our aim is to identify subtle changes in features that may indicate the proximity of the fault to failure. To achieve this, we introduce per-family features, complementing the event-based feature computation by characterizing families of seismic events. We use an unsupervised approach to categorize these families based on their feature similarities, grouping them into distinct categories revealing distinct patterns of seismicity. The high dimensional feature space of the categories prepares a floor to illuminate their joint evolution from normal/stable to critical/unstable state.

### Earthquakes with known preparatory phase

To test our approach, we employed three earthquake sequences, the first activating a major strike-slip plate boundary, the second occurring on a set of fragmented normal faults and the third occurred in a subduction zone. For all datasets, our workflow successfully identified categories of families (categories K4, L5 and I4, respectively) with distinct seismological features describing preparation phases[17]. It seems clear from the case studies here that individual features are more or less diagnostic (compare plots of features evolution, Supplementary Figs. 9 and S10); however, they are not as reliable as family-based features in revealing the critical category. The results for the Kahramanmaraş earthquake (Fig. 3a-c) align with previous studies[9,38,39]. However, our results provide a more refined view, identifying categories of event families signalling different stages of the preparatory process with different seismicity characteristics, some of them potentially representing the earthquake preparatory stage.

For the L'Aquila earthquake (Fig. 4), the presence of two distinct categories acting at different time (categories K4 and L5) during the preparatory phase is particularly interesting, as it aligns with previous studies that suggest a two-stage process: a phase lasting about few months characterized by low stress drop, slow migration, and more swarm-like behaviour, and a short activation phase with minor earthquakes being characterized by low seismic efficiency[60] and low $b$-value[61]. Our workflow captures changes in seismicity characteristics, with two dominant categories (L4-orange, L5-red). Comparing their topological properties shows that L4 (orange) tends to be more swarm-like, whereas L5 (red) exhibits mainshock–aftershock characteristics.

For the Iquique earthquake, the most critical categories emerge approximately two weeks before the mainshock. While previous studies have documented longer preparatory phases, spanning months to years of progressive weakening[32,44], our method primarily captures the final phase reflected in the seismic activity. These studies also report a combination of foreshock swarms and short slow-slip transients before and during the last two weeks. Accordingly, two categories are

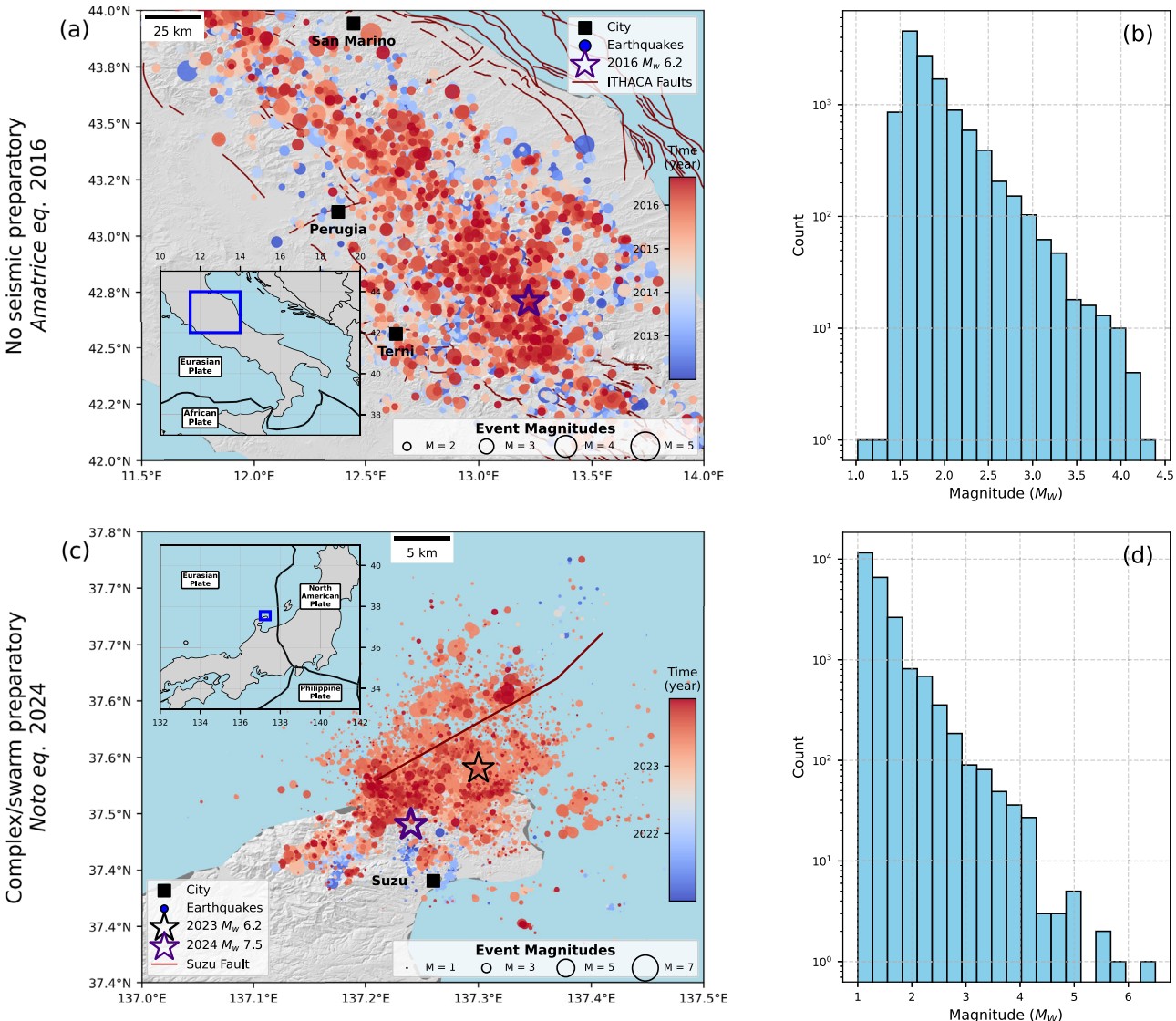

**Fig. 6 | Seismicity and magnitude distributions preceding two major earthquakes with unknown/complex preparatory phase. a, b** 4 years before 2016 MW 6.2 Amatrice and (**c, d**) 3 years before 2024 MW 7.5 Noto earthquakes. Events are shown in circles, sized by magnitude and color-coded by time. Mainshock epicenters are marked with stars. ITHACA ITaly HAzards from CApable faulting (Dataset is from the portal of the geological survey of Italy). Insets show location of maps.

observed within this time span (Fig. 5), with topological features indicating that the most critical category (I4) represents more swarm-like activity compared to I3.

### Earthquakes with unknown/complex preparatory phase

Our workflow relies solely on the seismicity catalog and its success depends directly on the presence of seismic activity during the preparatory phase. This means that our workflow will only succeed if earthquake interaction and clustering precede a mainshock. To illustrate this, we apply our workflow to the 2016 $M_w$ 6.2 Amatrice earthquake, also in central Italy, which was preceded by a well-documented, long-lasting period of quiescence[18]. (Fig. 6a, b). On the other hand, the presence of fluids is known to promote the occurrence of swarm-like seismicity[48], that statistically may still form events-families. We test the method for the 2024 $M_w$ 7.5 Noto earthquake (Fig. 6c, d) which represents a recent case displaying a long-lasting swarm activity preceding the mainshock, potentially representing a preparatory phase.

**Amatrice.** We use the seismic catalog data from the RAMONES service in central Italy[43]. To exclude aftershock activity from the 2009 $M_W$ 6.1 L'Aquila earthquake, we extract seismicity data starting from 2012. As expected, no category of seismicity, potentially signifying criticality, emerges in either time span for at least six months prior to the mainshock (Fig. 7). This may be due to either observational limitations or premonitory slip being dominantly aseismic. Sugan et al.[62] used a more enriched catalog and explained the unlocking process via progressive localization of the seismic activity at the fault edges. Figure 7 shows that in the final two months before the earthquake, category A1 (green) transitions to category A3 (yellow), indicating a change, particularly in the localization of clustered events (Fig. 7c); however, this pattern also appears in earlier months and years without leading to large earthquakes. This highlights that, even within a similar tectonic setting, all earthquakes may not share a similar initiation process[4].

**Noto.** The 2024 $M_W$ 7.5 Noto Peninsula earthquake struck northern Japan on January 1, 2024, with its hypocenter located within a region of intense swarm activity that began in late 2020 near Suzu, Ishikawa

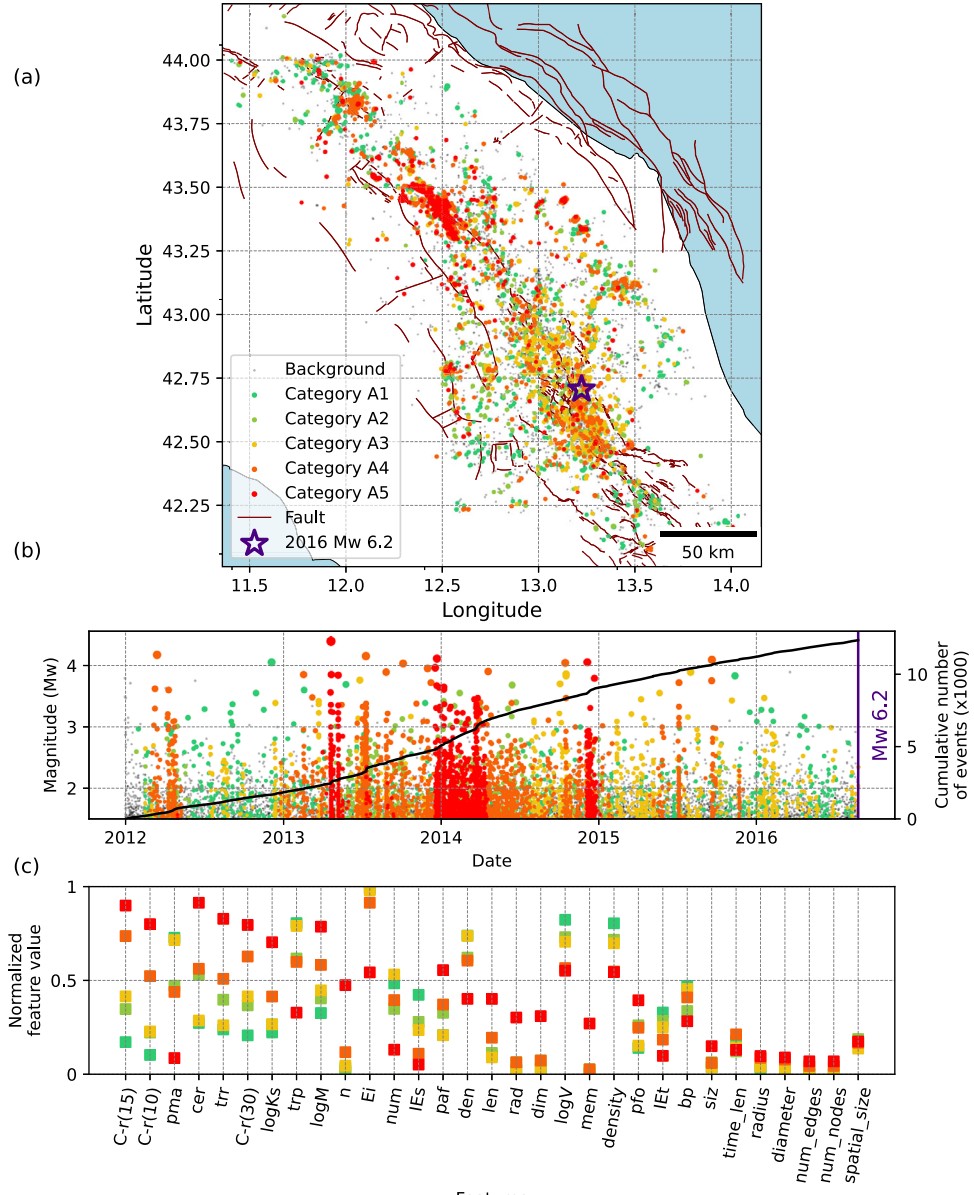

**Fig. 7 | Categorization of seismicity families prior to the 2016 MW 6.2 Amatrice earthquake. a** Map view showing the spatial distribution of event family members color-coded with their corresponding category. Background events are shown in grey. **b** Magnitude-time distribution of event family members, with the cumulative number of all (background and clustered) events represented by solid lines. **c** Feature values at the centroid of each category, sorted from highest to lowest separability. To improve visualization, we use an average value of each feature type over 4 spatial and temporal windows (2×2) and show only one value per feature type, leading to 30 (23+7) features. The colour scheme reflects the evolution of families, transitioning from a stable state (green) to a critical state (red). Description and explanations of individual features are provided in Table S1.

Province[63]. The mainshock displayed a reverse-faulting kinematics and occurred on a previously unrecognized fault, the Suzu Blind Fault (SBF), several kilometers deeper than the known Suzu-Oki active fault, which had been previously reactivated by the 2023 $M_W$ 6.2 event[63,64]. The mainshock was preceded by two years of westward-migrating swarm seismicity, potentially driven by fluids and aseismic slip. For this study, we use the relocated earthquake catalog, containing 23,630 events from 2021 to the mainshock with a magnitude of completeness $M_C = 1.1$[63].

Figure 8 shows the categorization results for this case. Two of the categories, (N2, olive green and N3, orange), exhibit similar localization and clustering characteristics preceding the 2023 $M_W$ 6.2 event. The immediate aftershock families of this relatively large event emerge as the most critical category (N4- red). According to Fig. 8b and also

depth-time separation of each category (see Supplementary Fig. 12), the seismicity pattern completely changes from dominant N2, N3 to N1 (green) category after this event. This is aligned with previous studies[64,65], showing that the 2023 $M_W$ 6.2 event ruptured a steep pre-existing reverse fault that had been lubricated by crustal fluids. Among all categories, category N3 displays distinct topological features (density, radius, and diameter of families) indicating that it captures most of the swarm activity. Finally, only category N1 (green) which is less critical than other categories persist until the 2024 mainshock.

We also compare them based on their feature values of the category centroid (see Supplementary Fig. 13). For all cases, we use features values for the most critical category, however, we use the category before the mainshock (A3) of the Amatrice and the main swarm-type category (N3) for the Noto case. The results

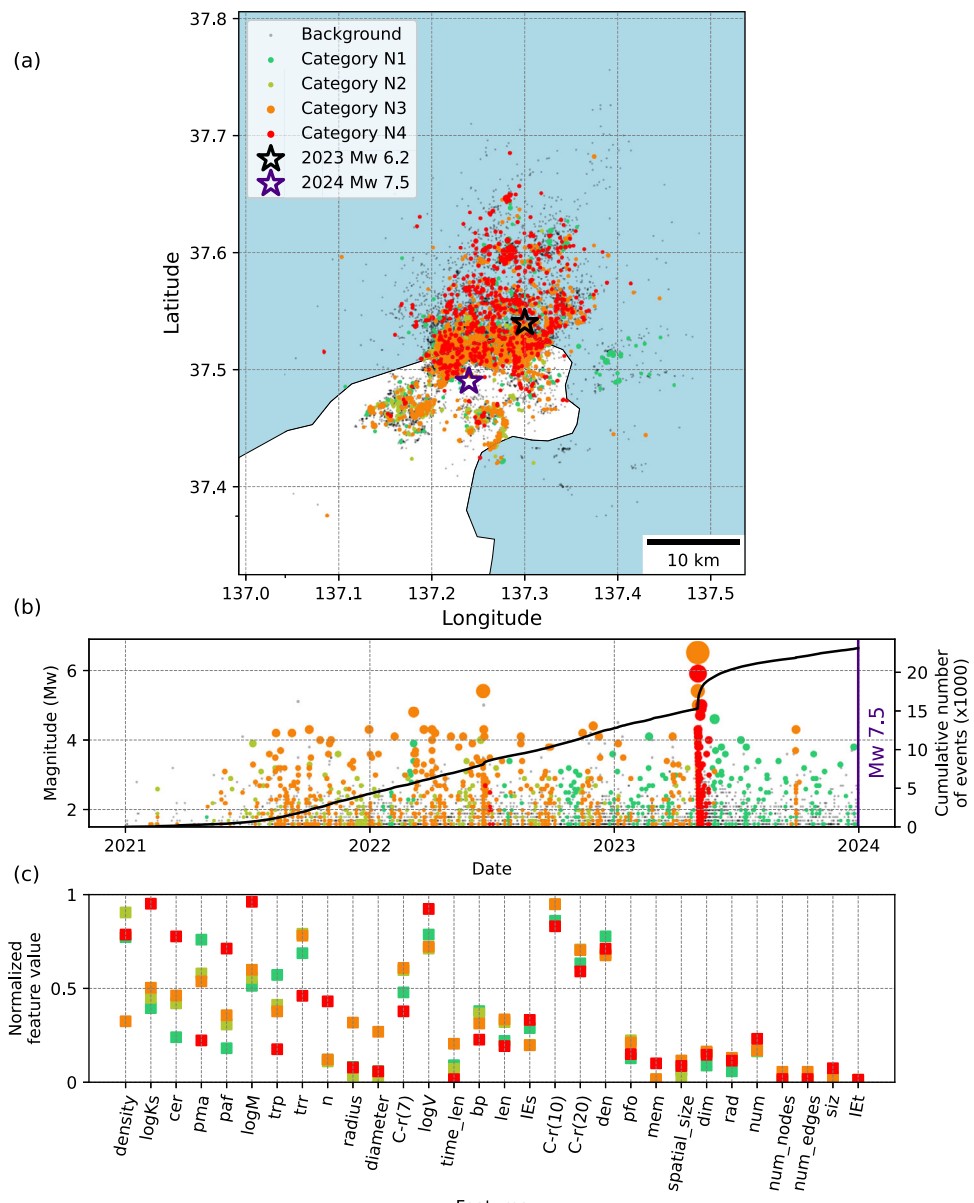

**Fig. 8 | Categorization of seismicity families prior to the 2024 MW 7.5 Noto earthquake. a** Map view showing the spatial distribution of event family members color-coded with their corresponding category. Background events are shown in grey. **b** Magnitude-time distribution of event family members, with the cumulative number of all (background and clustered) events represented by solid lines. **c** Feature values at the centroid of each category, sorted from highest to lowest separability. To improve visualization, we use an average value of each feature type over 4 spatial and temporal windows (2×2) and show only one value per feature type, leading to 30 (23+7) features. The colour scheme reflects the evolution of families, transitioning from a stable state (green) to a critical state (red). Description and explanations of individual features are provided in Table S1.

(Supplementary Fig. 13) show that the seismicity preceding the Kahramanmaraş and the Iquique earthquake exhibit the most critical feature values, followed by those preceding L'Aquila earthquakes. For example, these features include a lower event proximity (trp), higher event/moment rate (n, logM), high clustered events ratio (cer, pma) and generally higher spatial localization (C, siz). Topological features (especially lower density) also suggest that seismicity in the Iquique and Noto cases are more swarm-like than the other cases.

### Prospective applications for preparatory phase detection

Up to this point, we have presented retrospective results assuming prior knowledge of the timing of the large earthquake. Here, we now investigate the applicability of this method for preparatory phase detection (Fig. 9).

To this end, we consider a time window of previous data to fit the unsupervised model in each of our case studies (light blue areas in Fig. 9). We then move forward in time and, as each new seismicity family emerges, we employ the WCSS measure to evaluate whether the new family belongs to one of the categories identified during the fitting window. A significant increase in WCSS represents a distinct family with substantially different features, indicating the emergence of a new seismicity pattern. Based on the characteristics of the new family, an expert user can rank its criticality relative to previously identified categories.

For example, in the case of the 2023 Kahramanmaraş earthquake (Fig. 9a), we use catalog data from 2017 to 2019 to fit the unsupervised model, which identifies three categories. Advancing stepwise in time, each newly detected family is used to compute the difference in WCSS

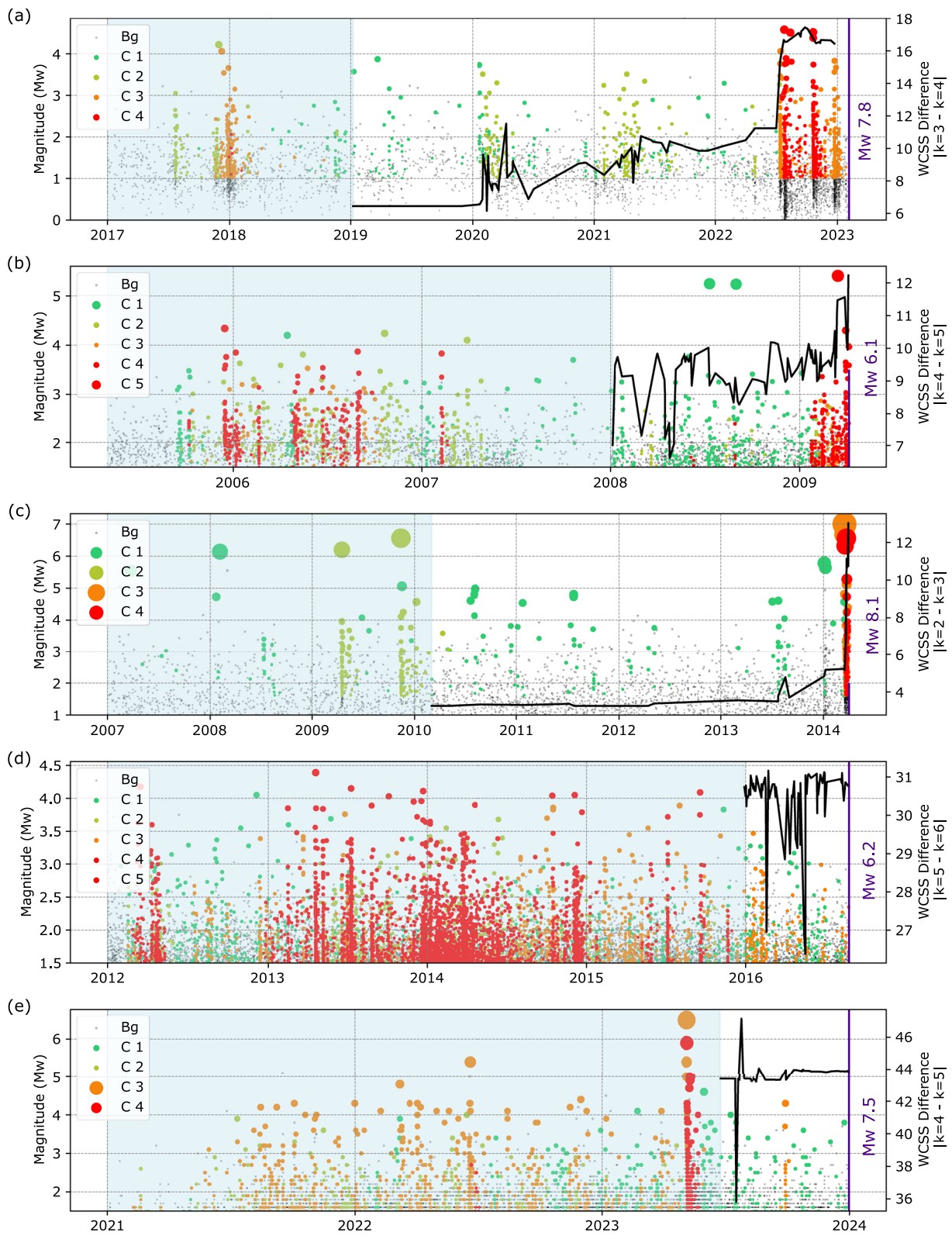

**Fig. 9 | Identification of preparatory phases preceding all major earthquakes. a** Kahramanmaraş, (**b**) L'Aquila, (**c**) Iquique, (**d**) Amatrice and (**e**) Noto earthquakes. The light blue shaded area indicates the time window used to fit the unsupervised model. The black curve represents the WCSS difference between for one additional category.

between models with the same number of categories (k = 3) and with one additional category (k = 4). A relatively small WCSS difference indicates that adding an extra category does not significantly improve separation of the families, implying that the new family is similar to those in the fitting set. In contrast, a larger WCSS difference suggests

that an additional category is required to more reliably separate the features of the seismicity. For the Kahramanmaraş earthquake, a pronounced increase in the WCSS difference (Fig. 9a) is observed from mid-2022 onward, coinciding with the onset of the preparatory phase of the earthquake. The features of the seismicity from the new

category (C4) indicate higher clustering productivity, stronger spatio-temporal localization, and increased strain release, emphasizing their elevated criticality relative to earlier categories.

Similar behaviour is observed in the other case studies. The WCSS difference reveals a transient phase approximately one month before the L'Aquila earthquake (Fig. 9b) and about two weeks before the Iquique earthquake in which an additional category of seismicity is required to explain the new features of the seismicity (Fig. 9c). In contrast, for two additional cases where the preparatory phase is not clearly reflected in the seismicity features, the WCSS difference plots do not show an increasing trend (Fig. 9d, e).

## Features reduction

As explained in the Methods section, a large number of features can be computed using various time and space windows, resulting in tens to hundreds of features. Many of these features are highly correlated to each other, meaning they convey similar information and may not all be necessary. This raises a key question that whether the results are producible with the most independent features and the most correlated features. We identify the most independent features through the following two-step process: 1. Feature grouping: All features are grouped into n classes, where each feature in a given class has a correlation of at least 70% (arbitrary threshold value) with at least one other feature in the same class. The number of classes is determined automatically based on the correlation threshold. This grouping ensures that features from different classes remain independent, as no two features from separate classes exceed the correlation threshold. 2. Representative feature selection: From each class, we select a single representative feature, the one with the highest average correlation to all other features within its class. The selected representative features, which are highly independent in the feature space, serve as input for the workflow to categorize seismic families.

For the 2023 Kahramanmaraş and 2009 L'Aquila earthquakes, the results (see Supplementary Fig. 14) reveal that the workflow generally produces similar outcomes when using the reduced set of independent features (compare Fig. 3 and Supplementary Fig. 14). As for the 2023 Kahramanmaraş earthquake, key independent features are b-value, clustering features (i.e., event proximity), and family features (i.e., time length of family and average number of families), whereas for the L'Aquila earthquake, they include b-value, correlation integral, moment rate, and family features (i.e., time length and diameter of family). This suggests that, while certain features may be case-specific, the most independent features tending to share common characteristics are primarily b-value (bp), temporal and spatial localization (trp, C-r), and family features (num, dia, len).

To identify the most correlated features, we apply the feature grouping approach described above to select sets of variables with pairwise correlations of at least 95%. We then categorize the seismic families using only the most correlated features. The results (Supplementary Fig. 15) demonstrate that a categorization can be consistently reproduced in both examples similar to those including all features using a subset of clustering features including event proximity (trp), clustered events ratio (cer), and proportion of mainshocks (pma).

A comparison with the most independent features further indicates that seismic interaction and clustering characteristics play the dominant role in the categorization, whereas some case-specific parameters, such as the b-value (bp), are required to achieve a more representative and comprehensive categorization.

## Limitations

Despite its advantages, this workflow has limitations, which we outline here. 1. Limited number of families: The dataset contains only 96 families for the Kahramanmaraş, 186 families for the L'Aquila and 42 families for the Iquique earthquake (Figs. S1-S6), even though the seismicity catalogs span 6 and 9 years, respectively. This small sample size makes it challenging to apply supervised learning algorithms. Even with unsupervised learning, a limited number of families may not provide enough variability to robustly identify categories that indicate the earthquake preparatory phase. Application to a longer dataset may help overcome this problem. 2. Determining the optimal number of categories: Although automated algorithms exist to estimate the optimal number of categories[35], expert interpretation is still required to establish a process-based understanding of how observed categories are related to earthquake generation. For example, Karimpouli et al[29]. demonstrated that applying a similar approach with three categories successfully revealed distinct seismicity stages, aligning well with the temporal evolution of stress in experimental data. When increasing the number of categories, however, this pattern became less distinguishable. 3. Feature normalization: Certain algorithms, such as K-means, require normalized input features, however; normalization is not feasible for all features used in this study. For example, while correlation integral is naturally constrained within the range of [0, 1], the event rate lacks a defined range, making normalization difficult and the workflow to be somewhat case-specific. Karimpouli et al.[29] suggested that simply removing such features may not significantly impact the results since correlated features may contain similar information. Our findings support this claim, as even when the feature set is substantially reduced, similar categories remain identifiable due to the high correlation among features.

## Methods

Figure 2 illustrates the proposed workflow in this work. Here, we explain more details about event-based features (Fig. 2, step 1.1), clustering/background analysis (Fig. 2, step 1.2, 2), per-family features (Fig. 2, step 3) and the K-means algorithm (Fig. 2, step 4).

### Event-based features

Given a seismicity catalog containing the time, location, and magnitude of events, one can compute various seismo-mechanical, statistical, and clustering features such as event rate, Gutenberg-Richter b-value, correlation integral and many others (Table S1). These features are commonly calculated at regular time intervals using a fixed time-step ($T_{step}$) and a time-window ($T_{win}$) (Supplementary Fig. S15). Unlike time-based methods, event-based feature computation directly incorporates the time and location of each event to define both time ($T_{win}$) and space ($S_{win}$) windows (Supplementary Fig. S16-S18). Feature values are assigned to individual events rather than fixed time intervals, and the process is repeated for each subsequent event. Following clustering analysis[12,34], seismic events are also classified as either background or clustered events to extract event families. This means that in a given time and space window, there are a set of families. In real time, families emerge that would include events that were previously not part of families. We use the average characteristics of all families such as number of families (or family rate), average members (events) of families and average size of families (in both time and space) as family features (Table S1) to assign into an event.

### Clustering/background analysis

Following previous studies[12,33,34], we use a data-driven approach to classify events as background or clustered using nearest-neighbour space-time-magnitude distances. The distance between $i$ and $j$ events is defined as[33]:

$$\eta_{ij} = \begin{cases} t_{ij}\left(r_{ij}\right)^{d_f} 10^{-bM_i} & t_{ij} > 0 \\ \infty & t_{ij} \leq 0 \end{cases} \quad (1)$$

where $t_{ij}$ and $r_{ij}$ are the time and space distance of two events, $d_f$ is the fractal dimension of event epicenters, $b$ is the b-value, and $M_i$ is the magnitude. Zaliapin et al.[34] defined the magnitude normalized time

and space components as:

$$T_{ij} = t_{ij} 10^{-qbM_i} \qquad (2)$$

$$R_{ij} = \left(r_{ij}\right)^{d_f} 10^{-(1-q)bM_i} \qquad (3)$$

where $\eta_{ij} = T_{ij} \times R_{ij}$ or in logarithmic scale $\log_{10}\eta_{ij} = \log_{10} T_{ij} + \log_{10} R_{ij}$ and usually $q$ is supposed to be 0.5. Subsequently, applying the Gaussian Mixture Model (GMM) approach on event distances ($\eta_{ij}$, two modes are detected and separated based on a threshold distance $\eta_0$ that equalizes the densities of the two estimated Gaussian modes known as background and clustered modes[12] (Fig. 2, step 1.2).

Connecting each event $i$ to its nearest neighbour (parent) event $j$, with a nearest-neighbour distance $\eta_{ij}$, results in a single cluster that contains all events connected to each other[33]. Removing all links that correspond to parent-offspring distances larger than the threshold distance $\eta_0$ creates a forest, including a collection of topological trees representing single events or families (Fig. 2, step 3).

### Family-based features

During Event-based computation, all features (including family features) are assigned to each event (Fig. 2, step 1.1) and, therefore, further analyses are based on features of individual events. We extend this workflow to the per-family feature computation, where features are assigned to one family (rather than per event). We reduce the high number of events into a small number of families (stars in Fig. 2, step 3), where all information within family members is summarized into per-family features. We use the mainshock location in time and space (i.e., the time of the largest magnitude event in the family, shown as bold circles in Fig. 2, step 2) to define each family in temporal and spatial dimensions. However, alternative definitions such as the first or last event time or a central location could also be used. For each family, we compute two sets of features: (a) Averaged Event-based features and (b) topological features. All features from both sets are assigned to each family as per-family features, which are subsequently used in an unsupervised learning step to identify distinct seismic family categories.

### K-means algorithm

In the last step (Fig. 2, step 4), we use the K-means algorithm[35] to categories per-family features. It is an iterative algorithm designed to partition data into $k$ clusters, where $k$ is a predetermined number based on a priori knowledge. Let $X$ be a set of $n$ data points in a $d$-dimensional space, $X \subset R^d$. The algorithm aims to select $k$ centroids, denoted as $C$, that minimize the inertia or squared error:

$$\sum_{x \in \mathbf{X}} \min_{c \in C} ||x - c||^2 \qquad (4)$$

The algorithm first calculates randomly initial k category centroids and assigns each data point to the nearest centroid. The centroids are then updated by computing the mean of the data points assigned to each category. This process continues iteratively until convergence, optimizing the within-category sum of squared distances. The convergence of the algorithm is heavily influenced by the initial selection of class centroids. To address this issue, Arthur & Vassilvitskii[66] proposed a K-means + + variation, which initializes the centroids to be distant from each other. This approach yields improved and faster results compared to random initialization.

### Data availability

The catalog data of each case used in this study is available online as follows. Kahramanmaraş: https://doi.org/10.5281/zenodo.15484902, Central Italy (L'Aquila and Amatrice): https://distav.unige.it/rsni/ramones.php, Iquique: https://doi.org/10.5880/GFZ.4.1.2023.004, and Noto: https://doi.org/10.5281/zenodo.12799164.

### Code availability

Codes are available in the GitHub repository: https://github.com/sadeghkarimpouli/Family_based_features, which is linked to Zenodo: https://doi.org/10.5281/zenodo.19254714

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

## Acknowledgements

SK, PMG and SNJ disclose support for the research of this work from the European Research Council (ERC) under the European Union's Horizon 2020 research and innovation program (Grant 101076119 for project QUAKEHUNTER). MP discloses support for the research of this work from the Italian PRIN projects PREPARED (2022ZHXWC9).

## Author contributions

SK: Conceptualization, Methodology, Validation, Visualization, Writing—original draft; PMG: Conceptualization, Supervision, Validation, Writing—review and editing; SNJ: Data curation, Validation, Writing—review and editing; MP: Conceptualization, Data curation, Validation, Writing—review and editing; DS: Data curation, Validation, Writing—review and editing; GK: Conceptualization, Validation, Writing—review and editing; GD: Conceptualization, Validation, Writing—review and editing; MB: Conceptualization, Validation, Writing—review and editing; GB: Conceptualization, Validation, Writing—review and editing.

## Funding

## Competing interests

The authors declare no competing interests.
