## [Transparent Peer Review file · Nature Communications]

Preparatory phase of large earthquakes illuminated by unsupervised categorization of earthquake catalog features

Corresponding Author: Dr Sadegh Karimpouli

Version 0:

Reviewer comments:

Reviewer #1

(Remarks to the Author)

The manuscript presents an innovative unsupervised machine learning (ML) framework aimed at identifying preparatory phases preceding large earthquakes through the analysis of seismicity patterns. The authors analyze three well-documented earthquake sequences: the 2023 Mw 7.8 Kahramanmaraş (Turkey) and 2009 Mw 6.1 L'Aquila (Italy) events, both with reported precursory activity, and contrast them with the 2016 Mw 6.1 Amatrice (Italy) earthquake, which lacked clear seismological precursors.

The methodology consists of three key stages: (1) defining seismic "families" based on spatio-temporal-magnitude proximity; (2) extracting 92–103 seismo-mechanical features per family (e.g., b-value, correlation integral, Kostrov strain, Energy Index); and (3) applying K-means clustering to categorize families into distinct seismicity stages, distinguishing between "stable" and "critical" states. The emergence of highly localized, high-strain families prior to mainshocks is interpreted as a potential signature of critical fault preparation.

This study represents a significant step toward data-driven, assumption-free analysis of earthquake preparation processes. It combines concepts from statistical seismology, laboratory fracture mechanics, and machine learning into a unified, reproducible framework. The authors commendably provide full access to data and code (via GitHub/Zenodo), and supplementary materials include detailed descriptions of feature computation and clustering diagnostics.

What I found very positive in this work is the following.

Novel and Conceptually Motivated Methodology

The shift from event-based to family-based analysis is a logical and valuable advancement. By grouping events into physically meaningful clusters, the method better captures interaction dynamics and spatial-temporal evolution of seismicity—offering a more holistic view than isolated event statistics.

Multidisciplinary Integration

The framework effectively bridges domains: it incorporates lab-derived precursory signals, statistical seismology metrics (e.g., b-value, correlation dimension), and topological/energetic indicators (e.g., Energy Index, strain release). This enhances the physical interpretability of identified patterns.

Case Study Design and Validation

The comparative design, i.e. analyzing two earthquakes with documented preparatory phases and one without, is a strong point. The method successfully identifies critical states before the Kahramanmaraş and L'Aquila events, while yielding no false positives for Amatrice, where no precursors are expected. This strengthens confidence in the method's discriminative capability.

Transparency and Reproducibility

The public availability of data, code, and extensive supplementary materials (including Tables S1–S2 and Figs. S1–S13) significantly enhances reproducibility and facilitates future benchmarking and extension by the community.

Evidence for Preparatory Phases

The study provides compelling evidence that certain seismic family characteristics, including high spatio-temporal localization, increased strain release, and energetic anomalies, systematically precede large earthquakes in some cases.

Scalable Workflow

The proposed pipeline offers a reproducible, scalable template for analyzing seismicity evolution, potentially adaptable to automated monitoring systems.

Physics-Informed ML

By selecting features with physical meaning, the approach avoids being purely data-driven “black box” modeling, enhancing interpretability and scientific value.

However, the paper has several limitations and critical concerns.

Conceptual and Methodological Constraints of Unsupervised Learning

While innovative, the unsupervised nature of the approach introduces key interpretive challenges:

Lack of Ground Truth: K-means clustering is agnostic to physical meaning. The assignment of “critical” labels to clusters is made a posteriori, based on temporal proximity to mainshocks and expert judgment. This risks confirmation bias.

Cluster Validation is Inadequate: The paper relies on the within-cluster sum of squares (WCSS) “elbow method” to determine the number of clusters (4–5), a subjective criterion. There is insufficient use of robust cluster validation metrics such as silhouette score, Davies-Bouldin index, or stability under bootstrapping permutation tests (e.g. Rousseeuw, P. J. 1987; Davies, D. L., & Bouldin, D. W. 1979).

Terminology Confusion: The use of “categorization” instead of “clustering” may mislead non-ML readers. Clearer adherence to standard ML terminology would improve clarity.

High Dimensionality and Feature Redundancy

The feature space (92–103 dimensions) is extremely large relative to the number of seismic families (96 for Kahramanmaraş, 186 for L’Aquila). Despite some evidence of feature independence (Fig. S7), this imbalance raises concerns about: The curse of dimensionality; Overfitting or spurious clustering; Sensitivity to normalization, particularly for scale-divergent features like event rate or energy; Dimensionality reduction (e.g., PCA, t-SNE, autoencoders) should be applied before clustering to improve robustness.

Small Sample Size and Generalizability

With only three case studies and limited family counts, the statistical power is low. The failure to detect a critical cluster in the Amatrice case—while consistent with the absence of precursors—also highlights the method’s potential lack of universality, even in similar tectonic settings. Broader validation across diverse tectonic environments (e.g., subduction zones, strike-slip, extensional regimes) is essential to assess generalizability.

Choice of Clustering Algorithm

K-means assumes spherical, isotropic, and equally sized clusters—assumptions unlikely to hold for complex spatio-temporal seismic patterns. More flexible, density-based (e.g., DBSCAN), probabilistic (e.g., Gaussian Mixture Models), or hierarchical clustering methods may better capture the underlying structure of seismic families.

Operational Applicability and Real-Time Feasibility

While the framework shows promise for retrospective “forensic” analysis, its utility for real-time forecasting remains speculative. Features are computed over aggregated time windows, which may delay detection of evolving critical states. The paper acknowledges this in the Discussion, but prospective testing or cross-validation under near-real-time conditions is absent.

Incomplete Sensitivity to Aseismic Processes

The reliance solely on seismic catalogs excludes aseismic deformation (e.g., slow slip events, creep), which may play a crucial role in earthquake preparation. Integrating geodetic (InSAR, GPS) or borehole strain data could improve the framework’s completeness and forecasting potential.

Based on the above issues, I would suggest the following issues for improvement

Enhance Clustering Robustness

Apply dimensionality reduction (e.g., PCA, UMAP) before clustering.

Test alternative clustering algorithms (e.g., DBSCAN, hierarchical, GMM) and compare results.

Report quantitative cluster validation metrics (silhouette, Davies-Bouldin, stability).

Expand Validation and Benchmarking

Include additional case studies, especially from different tectonic regimes and null cases (earthquakes without precursors).

Perform cross-validation across regions and time periods.

Consider semi-supervised approaches if expert-labeled “critical” periods are available.

Improve Feature Engineering

Reduce redundancy via feature selection or autoencoders.

Address normalization challenges for heterogeneous features.

Explore dynamic (time-evolving) feature representations.

Integrate Multimodal Data

Incorporate other kind of data, including geodetic, strain, or pore pressure data to account for aseismic processes.

Develop hybrid models that fuse seismic and non-seismic observables.

Assess Prospective Performance

Conduct retrospective prospective testing (e.g., sliding window analysis).

Evaluate detection lead times and false alarm rates.

Discuss practical implementation challenges for operational earthquake forecasting.

Conclusion

The paper presents a timely and thought-provoking study that advances the application of unsupervised machine learning in seismology. The family-based framework offers a fresh, physics-informed lens through which to examine earthquake preparation, and the results for Kahramanmaraş and L'Aquila are encouraging. However, the current methodology remains largely exploratory and interpretative, with limited predictive power and generalizability.

While the authors transparently acknowledge several limitations, the manuscript would benefit from major revisions, particularly in validation rigor, methodological diversity, and broader testing, before it can be considered a robust tool for operational forecasting. With these improvements, the work has the potential to become a foundational contribution to data-driven seismic hazard assessment.

Overall Recommendation: Major Revision

The study is innovative and well-structured, but requires substantial methodological strengthening and expanded validation to support its claims of identifying universal preparatory phases.

Minor points

Line 347. "... are also observed in las periods"; do you mean: "are also observed in last periods"?

Line 369. "AL5"; do you mean with L as pedix?

Line 394. How much do the classes change with change of the correlation percentage (e.g. with 75%)?

References

Davies, D. L., & Bouldin, D. W. (1979). A Cluster Separation Measure. *IEEE Transactions on Pattern Analysis and Machine Intelligence*, PAMI-1(2), 224–227

Rousseeuw, P. J. (1987). Silhouettes: A graphical aid to the interpretation and validation of cluster analysis. *Journal of Computational and Applied Mathematics*, 20, 53–65.

(Remarks on code availability)

Reviewer #2

(Remarks to the Author)

07/07/2025

Dear Editor,

The manuscript by Karimpouli et al. presents an unsupervised machine learning framework to identify precursory seismicity patterns preceding large earthquakes, based on the categorization of "event families" in a high-dimensional feature space. The method is applied to two well-studied sequences with documented preparatory phases (the 2023 Mw 7.8 Kahramanmaraş and the 2009 Mw 6.1 L'Aquila earthquakes) and a control case (the 2016 Mw 6.2 Amatrice earthquake) known for its lack of foreshocks. While the approach is conceptually interesting, the analysis is limited to only two positive cases and one negative example, all of which have been extensively studied in the literature. The method's capacity to detect truly novel or ambiguous precursory signals remains untested. Moreover, it is unclear whether the framework can distinguish between physically distinct phenomena such as foreshock-mainshock sequences and earthquake swarms, an essential capability given the growing recognition of diverse preparatory processes, exemplified by the recent 2024 Noto earthquake. The absence of testing across a broader range of seismic settings and catalog qualities severely restricts the generalizability of the findings. For these reasons, I do not recommend the manuscript for publication in *Nature Communications*. I provide below major and minor comments that may help strengthen the work.

Lack of discrimination between foreshocks and other transients. A critical concern in the interpretation of the results is whether the method is genuinely detecting foreshock sequences or merely identifying generic seismic anomalies, such as swarms. This distinction is not a minor detail, it lies at the heart of our ability to infer physical processes leading to large earthquakes. Understanding what differentiates a foreshock-mainshock sequence from an earthquake swarm is fundamental for both scientific insight and any operational forecasting potential. Recent examples, such as the 2024 Noto Peninsula earthquake in Japan, highlight the importance of this question: the sequence exhibited months of elevated seismicity prior to the mainshock, but debate persists as to whether this activity represented true foreshocks or a long-lived swarm with no direct connection to the mainshock rupture. The manuscript does not address this issue, nor does it attempt to test the method's capacity to distinguish between these two types of seismicity. Without such analysis, it remains unclear whether the approach is identifying processes directly related to earthquake nucleation, or simply flagging any increase in

spatiotemporal localization. This omission significantly limits the interpretability and potential utility of the results.

It is surprising that only three sequences are analyzed, especially considering that many other high-quality catalogs are publicly available and have been used in prior studies of foreshocks and seismic transients. For example, Ridgecrest (Beaucé et al., 2023; Shelly, 2020), Valparaíso (Ruiz et al., 2017; Moutote et al., 2022), Kumamoto (Kato et al., 2016), Noto (Peng et al., 2025), and Iquique (Kato et al., 2016; Schurr et al., 2020) all offer valuable test cases. Without broader testing across diverse settings, including swarms and mainshocks without precursors, the conclusions remain case-specific rather than generalizable.

Unclear physical interpretation of features. While the feature space is carefully constructed, the physical meaning of the identified categories remains ambiguous. For example, in the L'Aquila case, the Energy Index is included, but it is not used for Kahramanmaraş. The relative importance of such parameters, and their physical interpretation, are not sufficiently discussed. If the method is to be used for operational forecasting, understanding the underlying physics behind "critical" categories is essential.

Limited evaluation of false positives. To assess operational applicability, a more systematic analysis of false positives is required. The current single-case control (Amatrice) is insufficient. Additionally, the paper should compare its findings with competing interpretations, such as the unlocking process proposed by Sukan et al. (2023), particularly in the Kahramanmaraş case.

The methodological section is difficult to follow due to the large number of features and differences between the two sequences. A clear schematic diagram in the Supplementary Material illustrating the full workflow would greatly help reproducibility and understanding.

Although the feature-based family approach is well executed, the authors themselves acknowledge that the core methodology was previously developed and applied in the laboratory context. Given this, and the limited number of new applications, it is unclear whether the manuscript represents a sufficient advance to warrant publication in Nature Communications.

Minor Comments:

Figure 1: The spatial scales are inconsistent between panels a and b. A reference scale bar should be added to both for clarity.

Missing information: The paper does not show magnitude completeness (M_c) estimates or magnitude distributions for the catalogs used. These are essential for evaluating the robustness of any seismicity-based analysis.

L329–331: The statement that the method identifies "different stages of the preparatory process" is vague. The authors should clarify what physical processes these stages represent.

L343–345: The sentence acknowledges the need for testing on more sequences. Such validation should be a central component of the manuscript, not deferred to future work.

References:

- Beaucé, E., Poli, P., Waldhauser, F., Holtzman, B., & Scholz, C. (2023). Enhanced tidal sensitivity of seismicity before the 2019 magnitude 7.1 Ridgecrest, California earthquake. *Geophysical Research Letters*, 50(14), e2023GL104375.
- Kato, A., Fukuda, J. I., Kumazawa, T., & Nakagawa, S. (2016). Accelerated nucleation of the 2014 Iquique, Chile Mw 8.2 earthquake. *Scientific reports*, 6(1), 24792.
- Kato, A., Fukuda, J. I., Nakagawa, S., & Obara, K. (2016). Foreshock migration preceding the 2016 Mw 7.0 Kumamoto earthquake, Japan. *Geophysical Research Letters*, 43(17), 8945-8953.
- Moutote, L., Itoh, Y., Lengliné, O., Duputel, Z., & Socquet, A. (2023). Evidence of a transient aseismic slip driving the 2017 Valparaíso earthquake sequence, from foreshocks to aftershocks. *Journal of Geophysical Research: Solid Earth*, 128(9), e2023JB026603.
- Peng, Z., Lei, X., Wang, Q. Y., Wang, D., Mach, P., Yao, D., ... & Campillo, M. (2025). The evolution process between the earthquake swarm beneath the noto peninsula, central japan and the 2024 m 7.6 noto hanto earthquake sequence. *Earthquake Research Advances*, 5(1), 100332.
- Ruiz, S., Aden-Antoniow, F., Baez, J. C., Otarola, C., Potin, B., Del Campo, F., ... & Bernard, P. (2017). Nucleation phase and dynamic inversion of the Mw 6.9 Valparaíso 2017 earthquake in Central Chile. *Geophysical Research Letters*, 44(20), 10-290.
- Schurr, B., Moreno, M., Tréhu, A. M., Bedford, J., Kummerow, J., Li, S., & Oncken, O. (2020). Forming a Mogi doughnut in the years prior to and immediately before the 2014 Mw 8.1 Iquique, northern Chile, earthquake. *Geophysical Research Letters*, 47(16), e2020GL088351.
- Shelly, D. R. (2020). A high-resolution seismic catalog for the initial 2019 Ridgecrest earthquake sequence: Foreshocks, aftershocks, and faulting complexity. *Seismological Research Letters*, 91(4), 1971-1978.
- Sukan, M., Campanella, S., Chiaraluce, L., Michele, M., & Vuan, A. (2023). The unlocking process leading to the 2016 Central Italy seismic sequence. *Geophysical Research Letters*, 50(5), e2022GL101838.

(Remarks on code availability)

(Remarks to the Author)

Review of "Preparatory phase of large earthquakes illuminated by unsupervised categorization of earthquake catalog features" by Karimpouli et al.

The manuscript presents a timely study of the precursory phase of large magnitude earthquake incorporating techniques developed in the laboratory and extended here to geologic fault zones hosting large magnitude earthquakes. The technique presented applied seismicity catalog derived features to a clustering algorithm to determine spatial and temporal relationships of the prior events rupture sequence. The transition from the laboratory to the field is very useful. The authors state the limitations to this methodology in the discuss and a but additional points should be adding. Overall the manuscript is well written and with minor revisions considering the comments below, this manuscript will be well received by the community.

Comments:

1/ The technique developed previously and applied here makes many assumptions that are not defined in the limitations portion of the discussion. The stated limitation of "Determining the optimal number of categories" is significant. Without the historical context and time of known earthquake, how does one properly determine this value and identify when a transient is a false alarm (e.g., L271, L276-8)? This gets to the fundamental challenge of precursory behavior and more clearly stating this limitation would help the reader.

2/ The most important features are very similar to existing ideas of monitoring for temporal changes, e.g., b-value variations prior to large magnitude events. Here the b-value metric is identified as an important feature, but this has been previously shown with temporal statistics to be highly skewed to specific earthquakes and not a more general observations. Similar are the distance metrics and productivity rates that are found to be important. How does one use this information more broadly to identify a "true positive" transient?

3/ L339-43 The discussion here about false positives and negatives is not entirely clear. There is no metric presented that signals a "true positive" for the final precursory phase. This relates back to Q1. Determining the appropriate number of categories is subjective and defined here using historical context. It's stated that increasing the number of categories reduces the distinguishable patterns observed. What is the defined metric to determine a specific category will results in a large magnitude earthquake.

4/ The final paragraph of Discussions describes how this approach could be made operational. It would be helpful to add a sentence that the methodology would most likely be region specific and not a generalized solution. Describing how this could incorporate is beyond the scope here, but interesting nonetheless.

Christopher Johnson

(Remarks on code availability)

Version 1:

Reviewer comments:

Reviewer #1

(Remarks to the Author)

I am quite satisfied by the reply and the changes of the revised manuscript. There is only an exception: the present abstract does not actually reflect the changes made with the revision. In particular something should be said regarding the Amatrice-Norcia and Noto Earthquakes, and why they have been included in the study. The rest is fine to me.

(Remarks on code availability)

Reviewer #3

(Remarks to the Author)

Review of revisions for "Preparatory phase of large earthquakes illuminated by unsupervised categorization of earthquake catalog features" by Karimpouli et al.

The authors provided a mixed set of responses to the 3 reviewers that partially addressed the comments. Note, some were in the original version and did not require a detailed response. Specific to my previous comments, there is still the question of how one can determine that an identified category for a family be considered a transient precursory observation.

For example, L347 states "This suggests that the properties of clustered seismicity in the proximity of the mainshocks may be different from those at previous times." This the chicken or the egg conundrum, where knowing after categorizing earthquake activity up to the exact of the time of the mainshock allows the authors to claim a critical category exists prior to an earthquake as stated L25-8 "Their unsupervised categorization reflects distinct seismicity patterns. Results highlight the

occurrence of specific long-lasting families belonging to a critical category signalling an upcoming earthquake during the preparatory phase.”

The newly provided Iquique study most clearly highlights my previous comment about determining the optimal number of categories (unsupervised clusters). This point was raised by another reviewer which resulted in an extensive demonstration of unsupervised clustering metrics to determine the optimal number of clusters, with 3 of the 4 tests showing 2 clusters fit best, and only WCSS used in the analysis showing 4 clusters. The authors state in L128-9 “our framework effectively identifies a new category of seismicity before the mainshocks exhibiting anomalous properties relative to the other previous categories.” However, in the Iquique data shown in Figure 5, category I3 and I4 are identified 2 weeks prior to the mainshock while the data spans 7 years. This indicates that 362 weeks of data ($7\text{yrs} \times 52\text{wks/yr} - 2\text{wks} = 362\text{wks}$) contain no statistically significant information that defines a transient preceding a large earthquake and if evaluated without the mainshock should only contain 2 clusters. Without the historical context and timing of a known earthquake, how does one properly determine this metric to identify a transient? The response “We agree that the number of categories is a critical parameter; however, it can be considered as an assumption made during the training phase, which is mainly based on our knowledge about the region.” incorrectly defines the training phase.

Due note, these comments specifically ignore the details of spatial and temporal decisions made to calculate the features, that are most likely highly correlated as stated by the authors, and only focus on how one can successfully identify a precursory transient process from a non-precursory transient process using the algorithm presented to determine critical characteristics of a cluster family. If this information is presented in the results, then very clearly state this in the results and discussion so the reader is not confused.

The work is still valuable but the message remains muddled in the complexity of how the feature importance and families are presented with overlapping use of terminology that is adjusted for these issues. A clearer, more concise description of exactly how the authors decide a family exhibits a critical category defining a precursory signal is needed.

Some typos exist, this is not all of them.

L199 localisation -> localization

L342 (Figs. 3b, e) what is e?

Christopher Johnson

(Remarks on code availability)

Reviewer #4

(Remarks to the Author)

This manuscript proposes an unsupervised machine-learning approach to the analysis of earthquake catalog data, in which seismic events are grouped into families and analyzed using a set of derived features. While several questions and concerns remain, the proposed methodology represents an interesting and novel attempt at catalog-based analysis and therefore has potential value. On the other hand, the presentation and interpretation of the results place excessive emphasis on successful cases, raising concerns about the generality of the proposed approach. For these reasons, I consider that major revision is warranted.

The manuscript presents cases in which the proposed method does not work well, which I regard as scientifically honest. However, this balance is not reflected in the abstract. The abstract is written solely on the basis of successful examples and does not mention at all that the method fails in some cases. As a result, there is a clear discrepancy between the abstract and the main text. The abstract should acknowledge that the proposed method has limitations and does not perform well in all cases.

The abstract also suggests that the proposed approach may contribute to operational earthquake forecasting. Given the existence of cases in which the method does not work, this statement appears too strong at present. The wording should therefore be revised or more carefully qualified.

Moreover, even in the cases considered successful, the identification of a “critical category” is based on retrospective analyses of the entire catalog up to the mainshock, including long-term foreshock activity. This approach differs fundamentally from the conditions required for actual operational forecasting. If the authors had analyzed the catalog only up to a time preceding the onset of foreshock activity, and then demonstrated that subsequently occurring foreshocks can be identified as belonging to a category that is clearly distinct from all previously observed categories, the connection to operational earthquake forecasting would be much clearer. However, such an analysis is not presented in the current manuscript.

In the case of the Kahramanmaraş earthquake, false positives are observed, yet this issue is not discussed in sufficient detail. False positives are a critical factor in evaluating the practical usefulness of any forecasting-related method, and both quantitative and qualitative discussion of their occurrence and characteristics is necessary.

Figures 3c, 4c, and 5c present indicators associated with the critical category, but these indicators differ from one earthquake to another. The manuscript does not provide an interpretation of why different parameters appear to be important in different cases, making it difficult for readers to understand the underlying physical or statistical significance.

In addition, although multiple indicators associated with the critical category are presented, it remains unclear how these indicators are correlated with one another. While the manuscript includes a 'Features reduction' section, the analysis there primarily serves to justify the categorization consistency. It does not address the physical or statistical relationships among the specific indicators highlighted in the main results (e.g., Figs. 3c, 4c, 5c). To clarify the redundancy and independence of these features, the authors should provide a more detailed examination of their correlations. Such a discussion is crucial for understanding whether the observed changes in the critical category are driven by a few dominant physical processes or by a consensus of multiple independent parameters.

Finally, as a minor point, there is a typographical error in the last line of the abstract, where the phrase "forecasting. forecasting." is duplicated and should be corrected.

(Remarks on code availability)

Version 2:

Reviewer comments:

Reviewer #3

(Remarks to the Author)

Thank you for the details provided in the new Discussion section and Figure 9. This was additional work that now makes a much clearer presentation for the underlying metrics used in this analysis for identifying the transient changes before an event. The additional updates throughout are helpful and appreciated. I recommend this manuscript for publication.
Chris Johnson

(Remarks on code availability)

Reviewer #4

(Remarks to the Author)

I acknowledge that the authors have addressed my previous comments. I have no further comments and recommend the manuscript for publication.

(Remarks on code availability)

made.

Black: Associate editor and reviewers' comments.

Blue: Replies and discussions made by the authors.

Red: Changes made in the main text or supplementary materials.

Reviewer #1

The manuscript presents an innovative unsupervised machine learning (ML) framework aimed at identifying preparatory phases preceding large earthquakes through the analysis of seismicity patterns. The authors analyze three well-documented earthquake sequences: the 2023 Mw 7.8 Kahramanmaraş (Turkey) and 2009 Mw 6.1 L'Aquila (Italy) events, both with reported precursory activity, and contrast them with the 2016 Mw 6.1 Amatrice (Italy) earthquake, which lacked clear seismological precursors.

The methodology consists of three key stages: (1) defining seismic "families" based on spatio-temporal-magnitude proximity; (2) extracting 92–103 seismo-mechanical features per family (e.g., b-value, correlation integral, Kostrov strain, Energy Index); and (3) applying K-means clustering to categorize families into distinct seismicity stages, distinguishing between "stable" and "critical" states. The emergence of highly localized, high-strain families prior to mainshocks is interpreted as a potential signature of critical fault preparation.

This study represents a significant step toward data-driven, assumption-free analysis of earthquake preparation processes. It combines concepts from statistical seismology, laboratory fracture mechanics, and machine learning into a unified, reproducible framework. The authors commendably provide full access to data and code (via GitHub/Zenodo), and supplementary materials include detailed descriptions of feature computation and clustering diagnostics.

What I found very positive in this work is the following.

Novel and Conceptually Motivated Methodology

The shift from event-based to family-based analysis is a logical and valuable advancement. By grouping events into physically meaningful clusters, the method better captures interaction dynamics and spatial-temporal evolution of seismicity—offering a more holistic view than isolated event statistics.

Multidisciplinary Integration

The framework effectively bridges domains: it incorporates lab-derived precursory signals, statistical seismology metrics (e.g., b-value, correlation dimension), and topological/energetic indicators (e.g., Energy Index, strain release). This enhances the physical interpretability of identified patterns.

Case Study Design and Validation

The comparative design, i.e., analyzing two earthquakes with documented preparatory phases and one without, is a strong point. The method successfully identifies critical states before the

Kahramanmaraş and L’Aquila events, while yielding no false positives for Amatrice, where no precursors are expected. This strengthens confidence in the method’s discriminative capability.

Transparency and Reproducibility

The public availability of data, code, and extensive supplementary materials (including Tables S1–S2 and Figs. S1–S13) significantly enhances reproducibility and facilitates future benchmarking and extension by the community.

Evidence for Preparatory Phases

The study provides compelling evidence that certain seismic family characteristics, including high spatio-temporal localization, increased strain release, and energetic anomalies, systematically precede large earthquakes in some cases.

Scalable Workflow

The proposed pipeline offers a reproducible, scalable template for analyzing seismicity evolution, potentially adaptable to automated monitoring systems.

Physics-Informed ML

By selecting features with physical meaning, the approach avoids being purely data-driven “black box” modeling, enhancing interpretability and scientific value.

Respond: We appreciate the effort of the Reviewer on this manuscript, including a summary of its strengths. Among all the summarized points above, our most important goal was the presentation of a method, which is both scalable and explainable.

However, the paper has several limitations and critical concerns.

Conceptual and Methodological Constraints of Unsupervised Learning

While innovative, the unsupervised nature of the approach introduces key interpretive challenges:

Lack of Ground Truth: K-means clustering is agnostic to physical meaning. The assignment of “critical” labels to clusters is made a posteriori, based on temporal proximity to mainshocks and expert judgment. This risks confirmation bias.

Cluster Validation is Inadequate: The paper relies on the within-cluster sum of squares (WCSS) “elbow method” to determine the number of clusters (4–5), a subjective criterion. There is insufficient use of robust cluster validation metrics such as silhouette score, Davies-Bouldin index, or stability under bootstrapping permutation tests (e.g. Rousseeuw, P. J. 1987; Davies, D. L., & Bouldin, D. W. 1979).

Terminology Confusion: The use of “categorization” instead of “clustering” may mislead non-ML readers. Clearer adherence to standard ML terminology would improve clarity.

High Dimensionality and Feature Redundancy

The feature space (92–103 dimensions) is extremely large relative to the number of seismic families (96 for Kahramanmaraş, 186 for L’Aquila). Despite some evidence of feature independence (Fig. S7), this imbalance raises concerns about: The curse of dimensionality; Overfitting or spurious clustering; Sensitivity to normalization, particularly for scale-divergent

features like event rate or energy; Dimensionality reduction (e.g., PCA, t-SNE, autoencoders) should be applied before clustering to improve robustness.

Small Sample Size and Generalizability

With only three case studies and limited family counts, the statistical power is low. The failure to detect a critical cluster in the Amatrice case—while consistent with the absence of precursors—also highlights the method’s potential lack of universality, even in similar tectonic settings. Broader validation across diverse tectonic environments (e.g., subduction zones, strike-slip, extensional regimes) is essential to assess generalizability.

Choice of Clustering Algorithm

K-means assumes spherical, isotropic, and equally sized clusters—assumptions unlikely to hold for complex spatio-temporal seismic patterns. More flexible, density-based (e.g., DBSCAN), probabilistic (e.g., Gaussian Mixture Models), or hierarchical clustering methods may better capture the underlying structure of seismic families.

Operational Applicability and Real-Time Feasibility

While the framework shows promise for retrospective “forensic” analysis, its utility for real-time forecasting remains speculative. Features are computed over aggregated time windows, which may delay detection of evolving critical states. The paper acknowledges this in the Discussion, but prospective testing or cross-validation under near-real-time conditions is absent.

Incomplete Sensitivity to Aseismic Processes

The reliance solely on seismic catalogs excludes aseismic deformation (e.g., slow slip events, creep), which may play a crucial role in earthquake preparation. Integrating geodetic (InSAR, GPS) or borehole strain data could improve the framework’s completeness and forecasting potential.

Respond: Thank you for detailing the identified limitations of the method. Overall, we agree with the raised points, which are also generally applicable to most methods based on machine learning. In the following point-by-point comments, we tried to address all listed issues to the most possible extent.

I would suggest the following issues for improvement:

Enhance Clustering Robustness

1-1. Apply dimensionality reduction (e.g., PCA, UMAP) before clustering.

Respond: As the reviewer noted, one of the main strengths of our approach is that the resulting categories (clusters) remain interpretable, and preserving the explainability of the results is a key objective of our study. Although the feature set is relatively large, which is mainly due to the use of 4 sets of time and spatial windows, these features enable us to interpret each category. Applying dimensionality reduction techniques such as PCA or UMAP prior to categorization would compromise this interpretability, as the transformed features would no longer correspond directly to physically meaningful quantities. Hence, we do not see benefits in implementing this given the main purpose of our study.

1-2. Test alternative clustering algorithms (e.g., DBSCAN, hierarchical, GMM) and compare results.

Respond: We previously addressed this in our laboratory-scale study (see Fig. 3 in *Karimpouli et al., 2024*). In that work, we found that certain algorithms, such as Ward's hierarchical clustering, performed comparably to K-means. Nevertheless, we selected K-means because it is conceptually simpler, widely used, and computationally efficient. In response to the reviewer's suggestion, we have now repeated the analysis using several additional algorithms, including spectral clustering, 'Ward' hierarchical clustering, GMM, and DBSCAN. We also discuss the comparative results in the revised manuscript. Following are the changes that we made to both main text and supplementary materials:

Note that we also tested alternative clustering algorithms, including spectral clustering, 'Ward' hierarchical clustering, Gaussian Mixture Models (GMM), and Density-Based Spatial Clustering of Applications with Noise (DBSCAN). The results for the Kahramanmaraş earthquake are presented in Fig. S7-10, showing that only K-means and GMM produced coherent and physically meaningful categories that align well with the observed seismicity history in this region. Given the simplicity, interpretability, and computational efficiency of K-means, we employed this algorithm for all cases analyzed in this study.

Fig. S7. Categorization of seismicity families prior to the 2023 MW 7.8 Kahramanmaraş using **Spectral clustering** algorithm. Results show that 3 categories (K2-K4) are found in the preparatory phase of the 2023 mainshock and seismicity localization in 2018, while other families are accounted as one similar category (K1). This shows that this algorithm is not able to produce more details for less critical families.

Fig. S8. Categorization of seismicity families prior to the 2023 MW 7.8 Kahramanmaraş using ‘Ward’ hierarchical clustering algorithm. Compared to the K-means algorithm (Fig. 3), this algorithm separates less critical families more than more critical ones. For example, seismicity localization in 2018 is cauterized similar to the preparatory phase of the 2023 mainshock (K4), while only the later led to a mainshock.

Fig. S9. Categorization of seismicity families prior to the 2023 Mw 7.8 Kahramanmaraş using **Gaussian Mixture Model (GMM)** algorithm. These results are very similar to the results obtained by the K-means algorithm.

Fig. S10. Categorization of seismicity families prior to the 2023 MW 7.8 Kahramanmaraş using Density-Based Spatial Clustering of Applications with Noise (DBSCAN) algorithm. As it is illustrated,

DBSCAN mostly separates the families based on their topological features, where a critical family is assumed as noise to other families. We searched in a grid to find the optimum values for epsilon (maximum distance between two samples) and minimum number of samples in a neighborhood for a family to be considered as a core of a cluster.

Reference:

Karimpouli, S., Kwiatek, G., Martínez-Garzón, P., Dresen, G., & Bohnhoff, M. (2024). Unsupervised clustering of catalogue-driven features for characterizing temporal evolution of labquake stress. *Geophysical Journal International*, 237(2), 755-771.

1-3. Report quantitative cluster validation metrics (silhouette, Davies-Bouldin, stability).

Respond: In the initial version of the manuscript, we reported the quantitative metric of Within-Category Sum of Squares (WCSS) for different numbers of clusters (elbow plots) for both the Kahramanmaraş and L’Aquila earthquakes. In this revision, we have further examined several additional validation metrics, as shown in Fig. S11.

Fig. S11. Quantitative metrics for detecting number of categories (clusters) with (a) Within Category Sum of Square (WCSS) distance (elbow point is optimum), (b) silhouette score (the higher

the better), (c) Davies-Bouldin score (lower is better) and (d) Stability criterion (the higher the better).

We note that not all cluster evaluation criteria are equally suitable for this type of problem. Many standard metrics (e.g., silhouette, Davies–Bouldin, stability) are designed for generic clustering tasks and may not reliably reflect the physical relevance of categories in seismicity data. As illustrated in Fig. S11, some of these metrics suggest separating the data in only two clusters, which seems inconsistent with the seismological interpretation of the data. Therefore, we relied primarily on WCSS, which provides a meaningful and interpretable criterion consistent with the physical understanding of earthquake processes. Ultimately, the most reliable evaluation in this context involves an expert assessment of the resulting clusters in light of their seismological significance. The changes in the text are as follows:

Since the number of categories is not predefined, we determine the optimal number of categories using the within-category sum-of-squares (WCSS) criterion, which is a fundamental distance measure in K-means clustering. At the end, we find the optimal number based on interpretability of the categories. Our investigation shows that this is the most reliable criterion for this study (see Fig. S11). This yields four, five and four optimal categories for the Kahramanmaraş (K1-K4), L'Aquila (L1-L5) and Iquique (I1-I4) earthquakes, composed of different families, respectively.

Expand Validation and Benchmarking

1-4. Include additional case studies, especially from different tectonic regimes and null cases (earthquakes without precursors).

Respond: In the first version of the manuscript, we included two case studies from different tectonic regimes, the Kahramanmaraş earthquake representing a transform (strike-slip) setting and the L'Aquila earthquake representing an extensional (normal-fault) regime, as well as one null case (the Amatrice earthquake), where no clear precursory activity was observed. In this revised version, we have added two additional cases to further demonstrate the generalizability of the proposed method across different tectonic environments. For more details, please see our response to Comment 2-1 by reviewer #2.

1-5. Perform cross-validation across regions and time periods.

Respond: We find this comment somewhat general and open to interpretation. If, by “cross-validation across regions,” the reviewer refers to comparing the characteristics of critical families among different tectonic regions, this analysis was already included in the first version of the manuscript (previously Fig. S6). However, since we have now added two additional case studies, this figure has been updated accordingly (now Fig. S13). The new changes are as follow:

We also compare them based on their feature values of the category centroid (see Fig. S13). For all cases, we use features values for the most critical category, however, we use the category before the mainshock (A3) of the Amatrice and the main swarm-type category (N3) for the Noto case. The results (Fig. S13) show that the seismicity preceding the Kahramanmaraş and the Iquique earthquake exhibit the most critical feature values, followed by those preceding L’Aquila earthquakes. For example, these features include a lower event proximity (trp), higher event/moment rate (n, logM), high clustered events ratio (cer, pma) and generally higher spatial localization (C, siz). Topological features (specially lower density) also suggest that seismicity in the Iquique and Noto cases are more swarm-like than the other cases.

Fig. S13. Comparison of the features values in category centroids for all cases. For the three cases with known preparatory phases, i.e., Kahramanmaraş, L'Aquila and Iquique, we selected the most critical categories (K4, L5, and I4). We selected A3 for the Amatrice (as the category before the mainshock) and N3 for the Noto case (as the main swarm-type category).

1-6. Consider semi-supervised approaches if expert-labeled “critical” periods are available.

Respond: We agree that semi-supervised approaches can be valuable, particularly when the critical category is not clearly identified by unsupervised methods. However, in all our analyzed cases, the critical category was successfully distinguished from other categories through the per-family feature representation. This indicates that even if K-means were modified with constraints to perform semi-supervised clustering, the results would not significantly improve, as the critical category is already reliably captured by the unsupervised approach.

Improve Feature Engineering

1-7. Reduce redundancy via feature selection or autoencoders.

Respond: Feature selection was already performed in the first version of the manuscript. Details of this procedure and its impact on the results were previously described in the Discussion section and illustrated in Fig. S14.

1-8. Address normalization challenges for heterogeneous features.

Respond: Normalization of features was already discussed in the first version of the manuscript as a key potential limitation. Please refer to the Discussion section, where these challenges are addressed.

1-9. Explore dynamic (time-evolving) feature representations.

Respond: Features were visualized in the first version of the manuscript. Please refer to Figs. S16-17.

Integrate Multimodal Data

1-10. Incorporate other kinds of data, including geodetic, strain, or pore pressure data to account for aseismic processes.

Respond: We clearly note that the current method is explicitly based on seismic catalog data. While we agree that incorporating additional data types (e.g., geodetic, strain, or pore pressure measurements) could enhance the results, such an extension is beyond the scope of the present manuscript. It should also be noted that all of these data may not be available for all cases at the same spatial resolution.

1-11. Develop hybrid models that fuse seismic and non-seismic observables.

Respond: As with the previous comment, we acknowledge the potential benefits of hybrid models that integrate seismic and non-seismic observables. However, developing such models is beyond the scope of the current manuscript. Certainly, this will be an important topic for follow-up studies.

Assess Prospective Performance

1-12. Conduct retrospective prospective testing (e.g., sliding window analysis).

Respond: We agree that retrospective prospective testing is a valuable tool for evaluating the predictive stability of a model, particularly for time-dependent phenomena like earthquakes. However, our study does not aim to perform prediction, and the method presented here is not designed as a predictive model.

1-13. Evaluate detection lead times and false alarm rates.

Respond: As noted in response to the previous comment, our method does not aim to determine times to the mainshock nor to generate warnings. Our method focuses on identifying and characterizing seismicity patterns and not on forecasting.

1-14. Discuss practical implementation challenges for operational earthquake forecasting.

Respond: Practical implementation challenges are addressed in the last paragraph of the Discussion section, where we highlight the potentials of applying the method in real-time operational settings.

Minor points

Line 347. "... are also observed in las periods"; do you mean: "are also observed in last periods"?

Respond: Removed.

Line 369. "AL5"; do you mean with L as pedix?

Respond: Removed.

Line 394. How much do the classes change with change of the correlation percentage (e.g. with 75%)?

Respond: As described in the Discussion section, in general, increasing the correlation threshold leads to a similar or larger number of independent groups, which in turn results in a similar or higher number of independent features. For example, when comparing correlation thresholds of 70% and 75%, we observed the following: for the Kahramanmaraş earthquake, the number of reduced features increased by one, whereas for the L'Aquila earthquake, no additional features were selected (see below Figs. R1-2). These results indicate that the overall classification is relatively robust to moderate changes in the threshold correlation.

Fig. Review1. Results from categories of seismicity families based on reduced number of features prior to the 2023 Mw 7.8 Kahramanmaraş and 2009 Mw 6.1 L'Aquila earthquakes with a correlation threshold of 70 %.

Fig. Review2. Results from categories of seismicity families based on reduced number of features prior to the 2023 Mw 7.8 Kahramanmaraş and 2009 Mw 6.1 L'Aquila earthquakes with a correlation threshold of 75 %.

Reviewer #2

The manuscript by Karimpouli et al. presents an unsupervised machine learning framework to identify precursory seismicity patterns preceding large earthquakes, based on the categorization of “event families” in a high-dimensional feature space. The method is applied to two well-studied sequences with documented preparatory phases (the 2023 Mw 7.8 Kahramanmaraş and the 2009 Mw 6.1 L'Aquila earthquakes) and a control case (the 2016 Mw 6.2 Amatrice earthquake) known for its lack of foreshocks. While the approach is conceptually interesting, the analysis is limited to only two positive cases and one negative example, all of which have been extensively studied in the literature. The method's capacity to detect truly novel or ambiguous precursory signals remains untested. Moreover, it is unclear whether the framework can distinguish between physically distinct phenomena such as foreshock-mainshock sequences and earthquake swarms,

an essential capability given the growing recognition of diverse preparatory processes, exemplified by the recent 2024 Noto earthquake. The absence of testing across a broader range of seismic settings and catalog qualities severely restricts the generalizability of the findings. For these reasons, I do not recommend the manuscript for publication in Nature Communications. I provide below major and minor comments that may help strengthen the work.

Respond: We appreciate the reviewer's deep scientific insight and the important points raised. To clarify, our method is designed to detect various patterns of seismicity based on catalog-driven characteristics. As stated in the manuscript, it does not, by itself, determine the physical mechanisms behind these patterns. However, because the features are physically interpretable, experts can attribute mechanisms and/or criticality levels to each category, as we have done in the study. These interpretations are mainly based on general understanding of both laboratory and field observations and well-known earthquake models such as pre-slip, cascade, rate-dependent cascade-up and progressive localization models (*Kato and Ben-Zion 2021; Martínez-Garzón and Poli 2024; Peng and Lei 2025*). To test the capacity of the method for other cases, we have added two more cases that we discuss in the next comment (2-1), however; in response to this comment, we have now further explored the possibility of distinguishing swarm-like activities and mainshock–aftershock sequences. The following explanation has been added to the main text: To evaluate the occurrence of swarm-like activity, we use topological features of the families. Following Zaliapin & Ben-Zion (2013b), a topological average leaf depth can characterize the sequence, where the leaf depth counts the number of links from the first event to each leaf and a leaf is an event with no offspring. Similarly, we use additional graph-based properties to assess whether a family topology is chain-like (swarm) or umbrella-like (mainshock–aftershock or burst). For example, the density measures the number of connections relative to the maximum possible, where lower density indicates more swarm-like topology. Other measurements such as radius and diameter are derived from the eccentricity of the events defining the family. The radius/diameter are defined as the longest shortest path from a member to any other member, with the radius/diameter being the minimum/maximum eccentricity values. Accordingly, higher values of both metrics correspond to more swarm-like activity.

Using these measures, we can identify which categories exhibit more swarm-like characteristics in comparison with other categories. Notably, for the L'Aquila earthquake, the category immediately preceding the most critical category (L4-orange in Fig. 4) exhibits the lowest density and highest radius and diameter, indicating a more swarm-like topology. This finding aligns well with previous studies (Chiarabba et al., 2018; Sukan et al., 2014; Terakawa et al., 2010). This has been added to the Discussion section:

Our workflow captures changes in seismicity characteristics, with two dominant categories (L4-orange, L5-red). Comparing their topological properties shows that L4 (orange) tends to be more swarm-like, whereas L5 (red) exhibits mainshock–aftershock characteristics.

Finally, we have incorporated two additional case studies (Iquique and Noto earthquakes), which further support this analysis and are discussed in the main text and subsequent comments.

References:

- Chiarabba, C., De Gori, P., Cattaneo, M., Spallarossa, D. & Segou, M. Faults Geometry and the Role of Fluids in the 2016–2017 Central Italy Seismic Sequence. *Geophys Res Lett* 45, 6963–6971 (2018).
- Kato, A., Ben-Zion, Y., 2021. The generation of large earthquakes. *Nat. Rev. Earth Environ.* 2, 26–39.
- Martínez-Garzon, P., Poli, P., 2024. Cascade and preslip models oversimplify the complexity of earthquake preparation in nature. *Commun. Earth Environ.* 5, 120.
- Peng, Z., & Lei, X. (2025). Physical mechanisms of earthquake nucleation and foreshocks: Cascade triggering, aseismic slip, or fluid flows?. *Earthquake Research Advances*, 5(2), 100349.
- Sugan, M., Kato, A., Miyake, H., Nakagawa, S. & Vuan, A. The preparatory phase of the 2009 Mw 6.3 L'Aquila earthquake by improving the detection capability of low-magnitude foreshocks. *Geophys Res Lett* 41, 6137–6144 (2014).
- Terakawa, T., Zoporowski, A., Galvan, B. & Miller, S. A. High-pressure fluid at hypocentral depths in the L'Aquila region inferred from earthquake focal mechanisms. *Geology* 38, 995–998 (2010).

2-1. It is surprising that only three sequences are analyzed, especially considering that many other high-quality catalogs are publicly available and have been used in prior studies of foreshocks and seismic transients. For example, Ridgecrest (Beaucé et al., 2023; Shelly, 2020), Valparaíso (Ruiz et al., 2017; Moutote et al., 2022), Kumamoto (Kato et al., 2016), Noto (Peng et al., 2025), and Iquique (Kato et al., 2016; Schurr et al., 2020) all offer valuable test cases. Without broader testing across diverse settings, including swarms and mainshocks without precursors, the conclusions remain case-specific rather than generalizable.

Respond: We agree that it is difficult to assess the generalizability of the method based on only three cases. At the same time, we hope that the Reviewer can recognize that a single manuscript cannot cover all the cases cited. Even when covering all the cited cases, this would not ensure that our method is universally applicable. To partially address this important concern, we have added two additional case studies: the 2014 Mw 8.1 Iquique earthquake in Chile and the 2024 Mw 7.5 Noto earthquake in Japan. Our expanded dataset now includes:

A. Cases with well-documented preparatory phases in distinct tectonic settings:

1. The 2023 Mw 7.8 Kahramanmaraş earthquake along the East Anatolian Fault Zone transform boundary, Türkiye.
2. The 2009 Mw 6.1 L'Aquila earthquake in the extensional zone of central Italy.
3. The 2014 Mw 8.1 Iquique earthquake in the Chilean subduction zone.

B. Cases with unknown or complex preparatory phases:

4. The 2016 Mw 6.2 Amatrice earthquake in the central Apennines, Italy, where the preparatory phase is not evident in the seismicity catalog.

5. The 2024 M_W 7.5 Noto earthquake, Japan, where long-lasting swarm activity is recognized as the main preparatory mechanism.

As a result, we have re-organized the text and figures accordingly and, therefore, corresponding changes are found in several sections of the manuscript as follow:

Section: Results- The case studies

Iquique: The M_W 8.1 2014 Iquique is among the best-documented megathrust earthquakes in subduction zones, owing to extensive seismological and geodetic observations that captured its run-up, rupture onset, and the postseismic response. The mainshock was preceded by an extended preparatory phase, beginning about eight months earlier, characterized by interacting seismic and aseismic transients. During this period, bursts of seismicity and multiple slow-slip events were observed, the most prominent of them starting with an upper-plate M_W 6.7 foreshock two weeks before the mainshock (Ruiz et al., 2014; Socquet et al., 2017; Schurr et al., 2020; Boudin et al., 2022; Sippl et al., 2023a). Statistical analyses of the background seismicity and b -value patterns further revealed a years-long, progressive weakening of the future rupture area, setting the stage for the rupture of the main asperity (Schurr et al., 2014; Jara et al., 2017; Aden-Antoniow et al., 2020).

The Integrated Plate Boundary Observatory (IPOC) catalog is a semi-automatically compiled catalog covering 2007-2021 in Northern Chile (Sippl et al., 2018, 2023a, 2023b). It is built upon data from IPOC, the Centro Sismológico Nacional (CSN), GEOFON, and several temporary networks. For our analysis, we use the portion of the catalog spanning 2007 to 1 April 2014 (the day of the mainshock), covering the inter-plate and upper-plate seismicity around the eventual rupture zone. This subset contains 4134 events with a magnitude of completeness $M_c=1.7$, and includes numerous $M_W > 5$ earthquakes, among them two $M_W > 6$ events in 2008 and 2009 that did not culminate in a major rupture.

(a) Transform boundary
Kahramanmaraş eq. 2023

(c) Extensional zone
L'Aquila eq. 2009

(e) Subduction zone
Iquique eq. 2014

Fig. 1. Seismicity and magnitude distribution (a, b) 6 years before 2023 M_w 7.8 Kahramanmaraş, (c, d) 4 years before 2009 M_w 6.1 L'Aquila and (e, f) 7 years before 2014 M_w 8.1 Iquique earthquakes. Events are shown in circles, sized by magnitude and color-coded by time. Mainshock epicenters are marked with stars. Faults are shown in brown thin lines, while rupture along the EAFZ is shown in thick red line, in the case of Kahramanmaraş (a). ITHACA: ITaly HAZards from CApable faulting (Dataset is from the portal of the geological survey of Italy). Insets show location of maps.

Section: Results- Categorizing seismicity families using unsupervised learning

For the Iquique earthquake (Fig. 5), the two most critical categories (I3-orange and I4-red) are identified approximately two weeks prior to the mainshock (see the zoomed view in the top-right of Fig. 5). Similar to the other cases, these categories are characterized by high spatio-temporal localization (trp), elevated clustering activity (pma, cer), and high strain release (logK) as well as high event/moment rates (n, logM). In contrast, two $M_w > 6$ events in 2009 (I2-olive green) do not reach the same critical level, showing less prominent localization of seismicity..

Fig. 5. Categorization of seismicity families prior to the 2014 Mw 8.1 Iquique earthquake. The detailed description is similar to Fig. 3. Top-right is the zoom view of the last two weeks before the mainshock.

For the Iquique earthquake, highly critical categories emerge approximately two weeks before the mainshock. Although many studies reveal longer preparatory phases of months-long and even years-long progressive weakening (Ruiz et al., 2014; Schurr et al., 2014), our results only reveal the last phase expressed in seismic activity. Ruiz et al., (2014); Schurr et al., (2014) also explain a mixture of foreshock swarms and short slow-slip transients in the last two weeks. Accordingly, our method can identify two categories in this time span (zoom view in Fig. 5). The topological features of families included in the different categories reveal that category I4 represents more swarm-like activity compared to category I3.

Section: Discussion- Earthquakes with unknown/complex preparatory phase

On the other hand, the presence of fluids is known to promote the occurrence of swarm-like seismicity (Zaliapin and Ben-Zion, 2013b), that statistically may still form events-families. We test the method for the 2024 M_w 7.5 Noto earthquake (Fig. 6c, d) which represents a recent case displaying a long-lasting swarm activity preceding the mainshock, potentially representing a preparatory phase.

Fig. 6. Seismicity and magnitude distribution (a, b) 4 years before 2016 M_w 6.2 Amatrice and (c, d) 3 years before 2024 M_w 7.5 Noto earthquakes. The detailed descriptions are similar to Fig. 1.

Noto: The 2024 M_w 7.5 Noto Peninsula earthquake struck northern Japan on January 1, 2024, with its hypocenter located within a region of intense swarm activity that began in late 2020 near Suzu, Ishikawa Province. The mainshock displayed a reverse-faulting kinematics and occurred on a previously unrecognized fault, the Suzu Blind Fault (SBF), several kilometers deeper than the known Suzu-Oki active fault, which had been previously reactivated by the 2023 M_w 6.2 event (Yoshida et al., 2024; Kato, 2024). The mainshock was preceded by two years of westward-migrating swarm seismicity, potentially driven by fluids and aseismic slip. For this study, we use

the relocated earthquake catalog, containing 23,630 events from 2021 to the mainshock with a magnitude of completeness $M_c = 1.1$ (Yoshida et al., 2024, Zenodo).

Figure 8 shows the categorization results for this case. Two of the categories, (N2, olive green and N3, orange), exhibit similar localization and clustering characteristics preceding the 2023 M_w 6.2 event. The immediate aftershock families of this relatively large event emerge as the most critical category (N4- red). According to Fig. 8b and also depth-time separation of each category (see Fig. S12), the seismicity pattern completely changes from dominant N2, N3 to N1 (green) category after this event. This is aligned with previous studies (Kato 2024, Peng et al. 2025), showing that the 2023 M_w 6.2 event ruptured a steep pre-existing reverse fault that had been lubricated by crustal fluids. Among all categories, category N3 displays distinct topological features (density, radius, and diameter of families) indicating that it captures most of the swarm activity. Finally, only category N1 (green) which is less critical than other categories persist until the 2024 mainshock.

Fig. 8. Categorization of seismicity families prior to the 2024 M_w 7.5 Noto earthquake. The detailed description is similar to Fig. 3.

Supplementary material

Fig. S12. Depth distribution of four categories (a-d) in the 2024 Mw 7.5 Noto earthquake. The 2023 Mw 6.2 event shows up as the most critical category (d). The seismicity patterns change completely before (b, c) and after (d) this event, which reactivates the Suzu Blind Fault (SBF) and causing a new seismicity pattern separated from shallow to deep depths.

References:

- Aden-Antóniow F., Satriano C., Bernard P., Poiata N., Aissaoui E.M., Vilotte J.P., Frank W.B. Statistical analysis of the preparatory phase of the Mw 8.1 Iquique Earthquake, Chile *J. Geophys. Res.: Solid Earth*, 125 (2020), pp. 1-14.
- Boudin F., Bernard P., Meneses G., Vigny C., Olcay M., Tassara C., Boy J.P., Aissaoui E., Métois M., Satriano C., Esnault M.F., Necessian A., Vallée M., Vilotte J.P., Brunet C. Slow slip events precursory to the 2014 Iquique Earthquake, revisited with long-base tilt and GPS records *Geophys. J. Int.*, 228 (2022), pp. 2092-2121
- Jara J., Socquet A., Marsan D., Bouchon M. Long-term interactions between intermediate depth and shallow seismicity in north Chile subduction zone *Geophys. Res. Lett.*, 44 (2017), pp. 9283-9292.
- Kato, A. (2024). Implications of fault-valve behavior from immediate aftershocks following the 2023 Mj6. 5 earthquake beneath the Noto Peninsula, Central Japan. *Geophysical Research Letters*, 51(1), e2023GL106444.
- Peng, Z., Lei, X., Wang, Q. Y., Wang, D., Mach, P., Yao, D., ... & Campillo, M. (2025). The evolution process between the earthquake swarm beneath the Noto Peninsula, Central Japan and the 2024 M 7.6 Noto Hanto earthquake sequence. *Earthquake Research Advances*, 5(1), 100332.
- Ruiz S., Métois M., Fuenzalida A., Ruiz J., Leyton F., Grandin R., Vigny C., Madariaga R., Campos J. Intense foreshocks and a slow slip event preceded the 2014 Iquique Mw 8.1 earthquake *Science*, 345 (2014), pp. 1165-1170

Schurr B., Asch G., Hainzl S., Bedford J., Hoechner A., Palo M., Wang R., Moreno M., Bartsch M., Zhang Y., Oncken O., Tilmann F., Dahm T., Victor P., Barrientos S., Vilotte J.P. Gradual unlocking of plate boundary-controlled initiation of the 2014 Iquique earthquake *Nature*, 512 (2014), pp. 299-302.

Schurr B., Moreno M., Tréhu A.M., Bedford J., Kummerow J., Li S., Oncken O. Forming a mogi doughnut in the years prior to and immediately before the 2014 M8.1 Iquique, Northern Chile, *Earthquake Geophys. Res. Lett.*, 47 (2020)

Sippl C., Schurr B., Asch G., Kummerow J. Seismicity structure of the Northern Chile Forearc from >100,000 double-difference relocated hypocenters *J. Geophys. Res.*, 123 (2018), pp. 4063-4087

Sippl, C., Schurr, B., Münchmeyer, J., Barrientos, S., & Oncken, O. (2023). The Northern Chile forearc constrained by 15 years of permanent seismic monitoring. *Journal of South American Earth Sciences*, 126, 104326.

Sippl, Christian; Schurr, Bernd; Münchmeyer, Jannes; Barrientos, Sergio; Oncken, Onno (2023): Catalogue of Earthquake Hypocenters for Northern Chile from 2007-2021 using IPOC (plus auxiliary) seismic stations. GFZ Data Services.

Socquet A., Valdes J.P., Jara J., Cotton F., Walpersdorf A., Cotte N., Specht S., Ortega Culaciati F., Carrizo D., Norabuena E. An 8 month slow slip event triggers progressive nucleation of the 2014 Chile megathrust *Geophys. Res. Lett.*, 44 (2017), pp. 4046-4053

2-2. Unclear physical interpretation of features. While the feature space is carefully constructed, the physical meaning of the identified categories remains ambiguous. For example, in the L'Aquila case, the Energy Index is included, but it is not used for Kahramanmaraş. The relative importance of such parameters, and their physical interpretation, are not sufficiently discussed. If the method is to be used for operational forecasting, understanding the underlying physics behind "critical" categories is essential.

Respond: We agree that if the method were based on unknown or purely machine-learned features, physical interpretation would be challenging. However, this is one of the strengths of our approach. Each category is characterized by physical features, allowing an expert to interpret and label them based on the feature values. For example, we ranked categories for their criticality based on our laboratory and field observations of these features in addition to what we expect from physical earthquake models. While it is not possible to fully understand the underlying physics of each category solely from these results, the method provides a clear view of hidden seismicity patterns, which can help experts to compare the categories together and also to link them to the dominant physical processes in each region.

It is also important to note that ranking features by importance is context-dependent and depends on the specific goal. For example, in this study, we ranked features based on their separability among categories (see Figs. 3–5c and 7–8c) or identified most independent features (Fig. S14), but these rankings do not necessarily reflect their physical significance. We agree that incorporating additional information, such as aseismic processes, would be necessary for operational forecasting or for a deeper understanding of the physics behind "critical" categories. However, such extensions are beyond the scope of the current study.

2-3. Limited evaluation of false positives. To assess operational applicability, a more systematic analysis of false positives is required. The current single-case control (Amatrice) is insufficient.

Additionally, the paper should compare its findings with competing interpretations, such as the unlocking process proposed by Sukan et al. (2023), particularly in the Kahramanmaraş case.

Respond: Thank you for this comment. In this version, we have decided to not refer to the Amatrice case as a “false positive,” as this term may be misleading in the context of our study. Rather, the 2016 Amatrice represents a case with no critical seismicity pattern prior to the mainshock, within the temporal and spatial scope of our analysis. Similarly, the Noto case, which is added based on the reviewer’s suggestion, exhibits comparable behavior, with no critical seismicity pattern preceding the mainshock. In these instances, the method identifies no critical phase, which is a valid result based on the observed seismicity relative to the regional seismic history. More detailed studies, such as Sukan et al. (2023) on the 2016 Amatrice earthquake using enhanced catalogs, or including other types of data may provide further insight into the physical processes underlying these cases. We have also referenced the results of Sukan et al. (2023) in the revised manuscript.

2-4. The methodological section is difficult to follow due to the large number of features and differences between the two sequences. A clear schematic diagram in the Supplementary Material illustrating the full workflow would greatly help reproducibility and understanding.

Respond: We agree that completely documenting the large number of features is complicated, and we have now put additional effort on this endeavor. We have enhanced the flowchart (see Fig. 2) and added more detailed explanations to the text. The methodological section has been revised accordingly to allow readers to follow the workflow more easily. Most importantly, we have prepared a GitHub repository that provides all steps and resources necessary to reproduce the results. Those changes are reflected in the text as follow:

... We compute the **per-family** features as follows (Fig. 2):

Step 1.1. We compute and assign event-based features to each event (Fig. 2, also see Methods). To this end, we use 23 different features (Table S1). **These quantify general characteristics of the seismicity (event/moment rate, b-value), localisation in time, space, time-space and time-space-magnitude, earthquake interaction and properties of event families.** Based on two spatial and temporal windows (see Methods and Table S2), this leads to a pool of 92 features (23x2x2). However, for the L'Aquila earthquake, we also have access to the Energy Index ²⁷ and, therefore, we compute 96 features (24x2x2).

Step 1.2. The background and clustered events are identified based on nearest-neighbour distance and clustering analysis ³⁸ (Fig. 2). This provides the basis for the extraction of event families.

Step 2. A *family* is defined as a sequence of events connected by strong links based on nearest-neighbour distances below a threshold value (Fig. 2). We identify and extract 96, 186 and 44 different event families for the 2023 M_w 7.8 Kahramanmaraş, 2009 M_w 6.1 L'Aquila and 2014 M_w 8.1 Iquique earthquakes, respectively (see Figs. S1-S6).

Step 3. Per-family features are computed with two sub-sets of the features (Fig. 2): a) The average of all event-based features for each family (92, 96 and 92 features for Kahramanmaraş, L'Aquila and Iquique, respectively), and b) A set of 7 topological characteristics of the family such as the number of events, connection patterns, and other graph characteristics, that are independent of space and time windows (see bold features in Table S1). Therefore, a pool of 99, 103 and 99 features in total are computed for each family included in the catalogs of seismicity preceding the Kahramanmaraş, L'Aquila and Iquique earthquakes, respectively.

Step 4. We use per-family features as inputs for the K-means algorithm to categorize families in an unsupervised manner³⁴ (Fig. 2). This allows finding different seismicity patterns based on physics-informed features extracted from the seismicity catalog.

Fig. 2. Flowchart of the proposed method, showing a seismicity catalog in time and space, where the size of each event (circle) represents its magnitude. (Step 1.1) Event-based features (f_{E_i}) are computed based on the events inside spatial (S_{win}) and temporal (T_{win}) windows and are assigned to each event in its time (t_i) (see Methods and Table S1), **(Step 1.2)** all events are classified as either background (empty circles) or clustered seismicity (filled circles) following Zaliapin and Ben-Zion³⁹. **(Step 2)** We identify earthquake families from clustered seismicity by linking events according to

their nearest-neighbour distances after removing background seismicity. Each family consists of a mainshock (the largest-magnitude event within the family, shown with bold circles) and a foreshock–aftershock sequence (events occurring before–after the mainshock). Note that each event in the family is attributed a vector of n event-based features (f_E) from step 1.1. (Step 3) Per-family features (f_F) are derived for the mainshock time (represented by stars) by computing $\underline{f_{E_i}}$: the average values of all event-based features plus TF: topological features. (Step 4) K-means algorithm is used to categorize all families via unsupervised learning. Blue colored circles and stars show an example of i -th event and family. Green, orange and red colors show different categories of seismicity families.

We have additionally changed the ‘Methods’ section in the main text with this purpose.

2-5. Although the feature-based family approach is well executed, the authors themselves acknowledge that the core methodology was previously developed and applied in the laboratory context. Given this, and the limited number of new applications, it is unclear whether the manuscript represents a sufficient advance to warrant publication in Nature Communications.

Respond: We thank the reviewer for this important comment, and we dare to disagree for the following reasons: First, indeed we report that the core of the methodology was previously tested on a set of acoustic emission data from laboratory stick-slip experiments. However, the methodology is not entirely the same, and it has been heavily updated to rely more strongly on the identification and analysis of per-family features, which was not the case in laboratory experiments. Second, the key result of our analysis is not the generation of a new methodology. Rather, the main point is to effectively demonstrate that methodologies developed in the laboratory are possible to be upscaled to the field scale and show promising performance for earthquake sequences in nature. The implications of this are manifold. For example, as of to date earthquake prediction has been achieved in the laboratory under some circumstances, but obviously remains highly elusive in nature. The fact that methodologies developed for rock deformation experiments under controlled conditions can be effectively applied to nature represents a pioneer and important result, which, in our perspective, merits publication in Nature Communications which is why we chose the journal.

Minor Comments:

Figure 1: The spatial scales are inconsistent between panels a and b. A reference scale bar should be added to both for clarity.

Respond: Done.

Missing information: The paper does not show magnitude completeness (M_c) estimates or magnitude distributions for the catalogs used. These are essential for evaluating the robustness of any seismicity-based analysis.

Respond: Done.

L329–331: The statement that the method identifies “different stages of the preparatory process” is vague. The authors should clarify what physical processes these stages represent.

Respond: It is now rephrased to:

However, our results provide a more refined view, identifying categories of event families signalling different stages of the preparatory process with different seismicity characteristics, some of them potentially representing the earthquake preparatory stage.

L343–345: The sentence acknowledges the need for testing on more sequences. Such validation should be a central component of the manuscript, not deferred to future work.

Respond: Now we have added two more cases, see replies above.

Reviewer #3 (Dr. Christopher Johnson)

The manuscript presents a timely study of the precursory phase of large magnitude earthquakes incorporating techniques developed in the laboratory and extended here to geologic fault zones hosting large magnitude earthquakes. The technique applied seismicity catalog derived features to a clustering algorithm to determine spatial and temporal relationships of the prior events rupture sequence. The transition from the laboratory to the field is very useful. The authors state the limitations to this methodology in the discussion and additional points should be added. Overall the manuscript is well written and with minor revisions considering the comments below, this manuscript will be well received by the community.

Respond: Thank you, we appreciate your constructive and positive view of our manuscript.

3-1. The technique developed previously and applied here makes many assumptions that are not defined in the limitations portion of the discussion. The stated limitation of “Determining the optimal number of categories” is significant. Without the historical context and time of known earthquake, how does one properly determine this value and identify when a transient is a false alarm (e.g., L271, L276-8)? This gets to the fundamental challenge of precursory behavior and more clearly stating this limitation would help the reader.

Respond: We thank you for this comment. Since the method is based on unsupervised learning, it inherently relies on historical seismicity data for training (i.e., to find the categories). We agree that the number of categories is a critical parameter; however, it can be considered as an assumption made during the training phase, which is mainly based on our knowledge about the region. The choice of this parameter influences the categorization, but it reflects the structure of the underlying data rather than a predictive threshold for false alarms. With regard to the optimal

number of categories we tested different metrics, for example, for Kahramanmaraş earthquakes shown in the figure below:

Fig. S11. Quantitative metrics for detecting number of categories (clusters) with (a) Within Category Sum of Square (WCSS) distance (elbow point is optimum), (b) silhouette score (the higher, the better), (c) Davies-Bouldin score (the lower, the better) and (d) Stability criterion (the higher, the better).

We note that not all cluster evaluation criteria are equally suitable for this type of problem. Many standard metrics (e.g., silhouette, Davies–Bouldin, stability) are designed for generic clustering tasks and may not reliably reflect the physical relevance of categories in seismicity data. As illustrated in Fig. S11, some of these metrics suggest separating the data in only two clusters, which seems inconsistent with the seismological interpretation of the data. Therefore, we relied primarily on WCSS, which provides a meaningful and interpretable criterion consistent with the physical understanding of earthquake processes. Ultimately, the most reliable evaluation in this context involves an expert assessment of the resulting clusters in light of their seismological significance. The changes in the text are as follows:

Since the number of categories is not predefined, we determine the optimal number of categories using the within-category sum-of-squares (WCSS) criterion, which is a fundamental distance measure in K-means clustering. At the end, we find the optimal number based on interpretability of the categories. Our investigation shows that this is the most reliable criterion for this study (see Fig. S11). This yields four, five and four optimal categories for the Kahramanmaraş (K1-K4), L'Aquila (L1-L5) and Iquique (I1-I4) earthquakes, composed of different families, respectively.

3-2. The most important features are very similar to existing ideas of monitoring for temporal changes, e.g., b-value variations prior to large magnitude events. Here the b-value metric is identified as an important feature, but this has been previously shown with temporal statistics to be highly skewed to specific earthquakes and not a more general observation. Similar are the distance metrics and productivity rates that are found to be important. How does one use this information more broadly to identify a “true positive” transient?

Respond: This comment highlights the added value of our method. Unlike traditional approaches that rely on a few features such as b-value or event rate to describe seismicity, our method incorporates over 20 different features across four time and space windows, totaling approximately 100 features. This high-dimensional feature space allows the algorithm to uncover hidden patterns in seismicity that would be missed by single-feature analyses. Using this approach, one can compare the so-called “critical” categories across different cases to identify recurring trends. In our study, such comparisons revealed consistent patterns in the most critical categories across multiple earthquakes, providing a more robust framework for identifying significant seismicity transients. Although it was included in the first version of the manuscript, we now here rearrange it based on all five cases:

Importantly, our approach allows identifying a set of the most important features and their values distinguishing between family categories (Fig. 3-5c). These features are:

i) *Clustering features* such as the event proximity and the proportion of mainshocks to fore and aftershocks, which are observed to decrease, corresponding to the higher productivity of the sequences and the spatio-temporal localization.

ii) *The correlation integral*, which is observed to increase in agreement with the observed spatial point-localization of the event families.

iii) *Kostrov strain*, quantifying the enhanced seismic strain release, which is observed to increase during the preparatory phase.

iv) *3D volume and spatial interevent distance*, which are observed to decrease signifying again the higher event localization.

In addition, direct comparison of all cases is not straight forward, since some features have different ranges in different cases, however we add a new plot comparing all features of all cases in a normalized space:

We also compare them based on their feature values of the category centroid (see Fig. S13). For all cases, we use features values for the most critical category, however, we use the category before the mainshock (A3) of the Amatrice and the main swarm-type category (N3) for the Noto case. The results (Fig. S13) show that the seismicity preceding the Kahramanmaraş and the Iquique earthquake exhibit the most critical feature values, followed by those preceding L'Aquila earthquakes. For example, these features include a lower event proximity (trp), higher event/moment rate (n, logM), high clustered events ratio (cer, pma) and generally higher spatial localization (C, siz). Topological features (specially lower density) also suggest that seismicity in the Iquique and Noto cases are more swarm-like than the other cases.

Fig. S13. Comparison of the feature values in category centroids for all cases. For three cases with known preparatory phases, i.e., Kahramanmaraş, L'Aquila and Iquique, we selected the most critical categories (K4, L5, and I4). We selected A3 for the Amatrice (as the category before the mainshock) and N3 for the Noto case (as the main swarm-type category).

3-3. L339-43 The discussion here about false positives and negatives is not entirely clear. There is no metric presented that signals a “true positive” for the final precursory phase. This relates back to Q1. Determining the appropriate number of categories is subjective and defined here using historical context. It’s stated that increasing the number of categories reduces the distinguishable patterns observed. What is the defined metric to determine a specific category will result in a large magnitude earthquake.

Respond: We agree that the 'false/true positive/negative' were misleading for this manuscript, since, as you mentioned, there is no metric for that based on our results. Therefore, we decided to remove them from the updated manuscript. However, for the optimum number of categories please refer to the first comment (3-1).

3-4. The final paragraph of Discussions describes how this approach could be made operational. It would be helpful to add a sentence that the methodology would most likely be region specific and not a generalized solution. Describing how this could be incorporated is beyond the scope here, but interesting nonetheless.

Respond: We changed the corresponding sentence as follows.

Since this is a region-specific method, a long-term seismic catalog of a given region could be used to calibrate the model, allowing categorization of seismic families into well-defined categories.

Black: Reviewers comments.

Blue: Replies and discussions made by the authors.

Red: Changes made in the main text or supplementary materials.

Reviewer #1

I am quite satisfied by the reply and the changes of the revised manuscript. There is only an exception: the present abstract does not actually reflect the changes made with the revision. In particular something should be said regarding the Amatrice-Norcia and Noto Earthquakes, and why they have been included in the study. The rest is fine to me.

Respond: We appreciate the reviewer. The abstract is now changed according to this comment and the comments by another reviewer. As we are limited to 150 words, we have rephrased it accordingly.

Reviewer #3 (Dr. Christopher Johnson)

The authors provided a mixed set of responses to the 3 reviewers that partially addressed the comments. Note, some were in the original version and did not require a detailed response. Specific to my previous comments, there is still the question of how one can determine that an identified category for a family can be considered a transient precursory observation.

For example, L347 states “This suggests that the properties of clustered seismicity in the proximity of the mainshocks may be different from those at previous times.” This the chicken or the egg conundrum, where knowing after categorizing earthquake activity up to the exact of the time of the mainshock allows the authors to claim a critical category exists prior to an earthquake as stated L25-8 “Their unsupervised categorization reflects distinct seismicity patterns. Results highlight the occurrence of specific long-lasting families belonging to a critical category signalling an upcoming earthquake during the preparatory phase.”

The newly provided Iquique study most clearly highlights my previous comment about determining the optimal number of categories (unsupervised clusters). This point was raised by another reviewer which resulted in an extensive demonstration of unsupervised clustering metrics to determine the optimal number of clusters, with 3 of the 4 tests showing 2 clusters fit best, and only WCSS used in the analysis showing 4 clusters. The authors state in L128-9 “our framework effectively identifies a new category of seismicity before the mainshocks exhibiting anomalous properties relative to the other previous categories.” However, in the Iquique data shown in Figure 5, category I3 and I4 are identified 2 weeks prior to the mainshock while the data spans 7 years. This indicates that 362 weeks of data ($7\text{yrs} \times 52\text{wks/yr} - 2\text{wks} = 362\text{wks}$) contain no

statistically significant information that defines a transient preceding a large earthquake and if evaluated without the mainshock should only contain 2 clusters. Without the historical context and timing of a known earthquake, how does one properly determine this metric to identify a transient? The response “We agree that the number of categories is a critical parameter; however, it can be considered as an assumption made during the training phase, which is mainly based on our knowledge about the region.” incorrectly defines the training phase.

Due note, these comments specifically ignore the details of spatial and temporal decisions made to calculate the features, that are most likely highly correlated as stated by the authors, and only focus on how one can successfully identify a precursory transient process from a non-precursory transient process using the algorithm presented to determine critical characteristics of a cluster family. If this information is presented in the results, then very clearly state this in the results and discussion so the reader is not confused.

The work is still valuable but the message remains muddled in the complexity of how the feature importance and families are presented with overlapping use of terminology that is adjusted for these issues. A clearer, more concise description of exactly how the authors decide a family exhibits a critical category defining a precursory signal is needed.

Respond: Thank you for your critical, but constructive comments on the results in our manuscript. The main comment raised is fully valid and it was also noted by other reviewers. A main aspect to keep in mind in this discussion is that the path to reach the realization of automatic systems able to identify in real time a preparatory phase of a large earthquake is paved by many steps. In this work, we show that clustered seismicity before some large earthquakes has anomalous characteristics with respect to clustered seismicity in the interseismic period. We do not claim that this can be done automatically now, but it indicates there is a useful signal and suggests that there is a useful course to explore in the field of precursor research.

Given its importance, we have added a new subsection to the Discussion section entitled “Prospective applications for preparatory phase detection”. In this subsection, we demonstrate how the features of seismicity families, combined with unsupervised learning, can be used to distinguish a transient phase preceding a large earthquake. As clearly stated in the manuscript, we do not expect the proposed method to detect such a phase in cases where no preparatory phase is reflected in the seismicity features. In particular:

Prospective applications for preparatory phase detection

Up to this point, we have presented retrospective results assuming prior knowledge of the timing of the large earthquake. Here, we now investigate the applicability of this method for preparatory phase detection (Fig. 9).

To this end, we consider a time window of previous data to fit the unsupervised model in each of our case studies (light blue areas in Fig. 9). We then move forward in time and, as each new seismicity family emerges, we employ the WCSS measure to evaluate whether the new family belongs to one of the categories identified during the fitting window. A significant increase

in WCSS represents a distinct family with substantially different features, indicating the emergence of a new seismicity pattern. Based on the characteristics of the new family, an expert user can rank its criticality relative to previously identified categories.

For example, in the case of the 2023 Kahramanmaraş earthquake (Fig. 9a), we use catalog data from 2017 to 2019 to fit the unsupervised model, which identifies three categories. Advancing stepwise in time, each newly detected family is used to compute the difference in WCSS between models with the same number of categories ($k = 3$) and with one additional category ($k = 4$). A relatively small WCSS difference indicates that adding an extra category does not significantly improve separation of the families, implying that the new family is similar to those in the fitting set. In contrast, a larger WCSS difference suggests that an additional category is required to more reliably separate the features of the seismicity. For the Kahramanmaraş earthquake, a pronounced increase in the WCSS difference (Fig. 9a) is observed from mid-2022 onward, coinciding with the onset of the preparatory phase of the earthquake. The features of the seismicity from the new category (C4) indicate higher clustering productivity, stronger spatio-temporal localization, and increased strain release, emphasizing their elevated criticality relative to earlier categories.

Similar behavior is observed in the other case studies. The WCSS difference reveals a transient phase approximately one month before the L'Aquila earthquake (Fig. 9b) and about two weeks before the Iquique earthquake in which an additional category of seismicity is required to explain the new features of the seismicity (Fig. 9c). In contrast, for two additional cases where the preparatory phase is not clearly reflected in the seismicity features, the WCSS difference plots do not show an increasing trend (Fig. 9d, e).

Fig. 9. Identification of the preparatory phase for the (a) Kahramanmaraş, (b) L'Aquila, (c) Iquique, (d) Amatrice and (e) Noto earthquakes. The light blue shaded area indicates the time

window used to fit the unsupervised model. The black curve represents the WCSS difference between for one additional category.

Reviewer #4

This manuscript proposes an unsupervised machine-learning approach to the analysis of earthquake catalog data, in which seismic events are grouped into families and analyzed using a set of derived features. While several questions and concerns remain, the proposed methodology represents an interesting and novel attempt at catalog-based analysis and therefore has potential value. On the other hand, the presentation and interpretation of the results place excessive emphasis on successful cases, raising concerns about the generality of the proposed approach. For these reasons, I consider that major revision is warranted.

Respond: Thank you for the valuable comments and the constructive evaluation.

The manuscript presents cases in which the proposed method does not work well, which I regard as scientifically honest. However, this balance is not reflected in the abstract. The abstract is written solely on the basis of successful examples and does not mention at all that the method fails in some cases. As a result, there is a clear discrepancy between the abstract and the main text. The abstract should acknowledge that the proposed method has limitations and does not perform well in all cases.

The abstract also suggests that the proposed approach may contribute to operational earthquake forecasting. Given the existence of cases in which the method does not work, this statement appears too strong at present. The wording should therefore be revised or more carefully qualified.

Respond: Thank you for this comment. **The abstract is now changed** according to this comment and the comments by the other reviewer, taking into account that we are limited to 150 words. We also note that, to address the concerns from Reviewer 4, we have now added a discussion section dealing with the topic of prospective for preparatory phase detection, and hence we have decided to leave that part in the abstract.

Moreover, even in the cases considered successful, the identification of a “critical category” is based on retrospective analyses of the entire catalog up to the mainshock, including long-term foreshock activity. This approach differs fundamentally from the conditions required for actual operational forecasting. If the authors had analyzed the catalog only up to a time preceding the onset of foreshock activity, and then demonstrated that subsequently occurring foreshocks can be identified as belonging to a category that is clearly distinct from all previously observed categories, the connection to operational earthquake forecasting would be much clearer. However, such an analysis is not presented in the current manuscript.

Respond: We agree that the potential for operational forecasting is one of the most important aspects of our method, as also emphasized by the other reviewers. In light of this comment, and in response to a similar remark by Reviewer 3, we have added a new subsection to the Discussion section entitled “Prospective applications for preparatory phase detection”. To avoid repetition, we kindly refer to our detailed **response to Reviewer 3**.

In the case of the Kahramanmaraş earthquake, false positives are observed, yet this issue is not discussed in sufficient detail. False positives are a critical factor in evaluating the practical usefulness of any forecasting-related method, and both quantitative and qualitative discussion of their occurrence and characteristics is necessary.

Respond: If we have correctly understood, the Reviewer is referring to the time period in late 2017-early 2018 that shows the category K3 (Fig. 3 at 2017-2018). Even though this could be interpreted as a false positive, were this to be an operational approach, we would likely require that a critical category should persist over time for it to be translated as a criticality in an operational earthquake forecasting sense (for example, as volcano early warning systems). In the long term plan of developing a forecasting system, we agree with the reviewer that further systematic studies on large datasets must also investigate the issue of false positives. We included a comment about this in the discussion.

Figures 3c, 4c, and 5c present indicators associated with the critical category, but these indicators differ from one earthquake to another. The manuscript does not provide an interpretation of why different parameters appear to be important in different cases, making it difficult for readers to understand the underlying physical or statistical significance.

Respond: Our results highlight that earthquakes do not share identical initiation processes, and therefore it should not be expected that the same parameters should be used to characterize all cases. We agree that it is not straightforward to interpret what is going on behind the scenes with the unsupervised categorization, but we observe that the controlling features: (i) clustering features, (ii) correlation integral, (iii) kostrov strain, and (iv) 3D volume and spatial interevent distance) are consistent with papers defining stress criticality (Eneva and Ben-Zion, 1997), the progressive localization model (Kato and Ben-Zion 2020) and overall features of the seismicity through the laboratory seismic cycle (Kwiatek et al. 2024). The unsupervised categorization method captures that process and explains why it falls short in more complex cases (Noto) or cases where quiescence precedes the mainshock (Amatrice).

In addition, although multiple indicators associated with the critical category are presented, it remains unclear how these indicators are correlated with one another. While the manuscript includes a 'Features reduction' section, the analysis there primarily serves to justify the categorization consistency. It does not address the physical or statistical relationships among the specific indicators highlighted in the main results (e.g., Figs. 3c, 4c, 5c). To clarify the redundancy

and independence of these features, the authors should provide a more detailed examination of their correlations. Such a discussion is crucial for understanding whether the observed changes in the critical category are driven by a few dominant physical processes or by a consensus of multiple independent parameters.

Respond: We appreciate the comment raised by the reviewer. Due to the large number of features, we need an automated approach to find such relationships among features from the correlation matrix. In the previous version, we extracted the most independent features, however, based on this comment, we now find the most correlated features and reproduce the results. Following is the discussion and new figures we add to the main text and supplementary material:

To identify the most correlated features, we apply the feature grouping approach described above to select sets of variables with pairwise correlations of at least 95%. We then categorize the seismic families using only the most correlated features. The results (Fig. S15) demonstrate that a categorization can be consistently reproduced in both examples similar to those including all features using a subset of clustering features including event proximity (*trp*), clustered events ratio (*cer*), and proportion of mainshocks (*pma*).

A comparison with the most independent features further indicates that seismic interaction and clustering characteristics play the dominant role in the categorization, whereas some case-specific parameters, such as the *b*-value (*bp*), are required to achieve a more representative and comprehensive categorization.

Fig. S14. Results from categories of seismicity families based on the most independent features prior to the 2023 M_W 7.8 Kahramanmaraş and 2009 M_W 6.1 L'Aquila earthquakes. (a, c) Magnitude-time distribution of event family members, with the cumulative number of (all background and clustered) events represented by solid lines. (b, d) Feature values at the centroid

of each category, sorted from highest to lowest separability. The color scheme reflects the evolution of families, transitioning from a stable state (green) to a critical state (red). Description and explanations of individual features is provided in Table S1. ‘T’ and ‘S’ represent time and space windows in ‘day’ and ‘km’.

Fig. S15. Results from categories of seismicity families based on the most correlated features prior to the 2023 M_w 7.8 Kahramanmaraş and 2009 M_w 6.1 L'Aquila earthquakes. (a, c) Magnitude-time distribution of event family members, with the cumulative number of (all background and clustered) events represented by solid lines. (b, d) Feature values at the centroid of each category, sorted from highest to lowest separability. The color scheme reflects the evolution of families, transitioning from a stable state (green) to a critical state (red). Description and explanations of individual features is provided in Table S1. ‘T’ and ‘S’ are time and space windows in ‘day’ and ‘km’.